# VARIANCE REDUCED DOMAIN RANDOMIZATION FOR POLICY GRADIENT

## ABSTRACT

By introducing randomness on environment parameters that fundamentally affect the dynamics, domain randomization (DR) imposes diversity to the policy trained by deep reinforcement learning, and thus improves its capability of generalization. The randomization of environments, however, introduces another source of variability for the estimate of policy gradients, in addition to the already high variance due to trajectory sampling. Therefore, with standard state-dependent baselines, the policy gradient methods may still suffer high variance, causing low sample efficiency during the training of DR. In this paper, we theoretically derive a bias-free and state/environment-dependent optimal baseline for DR, and analytically show its ability to achieve further variance reduction over the standard constant and state-dependent baselines for DR. We further propose a variance reduced domain randomization (VRDR) approach for policy gradient methods, to strike a tradeoff between the variance reduction and computational complexity in practice. By dividing the entire space of environments into some subspaces and estimating the state/subspace-dependent baseline, VRDR enjoys a theoretical guarantee of faster convergence than the state-dependent baseline. We conduct empirical evaluations on six robot control tasks with randomized dynamics. The results demonstrate that VRDR can consistently accelerate the convergence of policy training in all tasks, and achieve even higher rewards in some specific tasks.

## 1 INTRODUCTION

Deep reinforcement learning (DRL) has achieved an impressive success on complex sequential decision-making tasks, such as playing video games at top human-level (Ye et al., 2020), continuous robot manipulation (Agarwal et al., 2021) and traffic signal control (Egeaa et al., 2021). For an extensive exploration, DRL requires a massive amount of samples to train the policy, which is usually done in a simulator constructed for the task. Due to the lack of diversity, however, policy trained by DRL tends to overfit to a specific training environment (Cobbe et al., 2019), causing possibly a severe performance degradation (i.e., reality gap) when the policy learned in simulator is transferred to the real deployment (Peng et al., 2018; Kang et al., 2019). To close this gap, domain randomization (DR) has been proposed to randomize the environment parameters that may vary the dynamics and observation in simulator, imposing diversity on the trained policy (Tobin et al., 2017; Jiang et al., 2021). Recent success in robot control has shown that DR enables the trained policy to generalize in real deployment, even with presence of fundamental modeling errors in the simulator (Muratore et al., 2021; Xie et al., 2020). DR has also demonstrated its capability of robustness to contend with extremely rare environments (Muratore et al., 2021).

Direct applications of DR on policy gradient methods can be thought as policy optimization on multiple randomly generated environments, which incurs additional variability of the observed data. Therefore, a critical challenge is the low sample efficiency due to the extremely high variance of the gradient estimator, where the variability not only stems from the gradient approximation of expected return from a batch of sampled trajectories (Greensmith et al., 2004), but is also imposed by the randomization of environment parameters. In standard DRL, a commonly used method to reduce variance is to construct a bias-free and action-independent baseline that is subtracted from the expected return (Sutton & Barto, 2018). For example, a typical choice of baseline is the state value function of a policy. When directly applied to DR, as proposed by Andrychowicz et al. (2020), it is equivalent to learning a state value function that predicts the expected return over all possible

environments. Though remaining unbiased, such a state-dependent baseline may be a poor choice in DR, since the additional variability of randomized environments is not taken into account.

In this paper, we aim to address the high variance issue of policy gradient methods for domain randomization, with a particular focus on reducing the additional variance imposed by randomization of environments. Our key insight is that the additional information on varying environments can be incorporated into the baseline to further reduce this variance. Theoretically, we derive the optimal state/environment-dependent baseline, and demonstrate that it improves consistently the variance reduction over baselines that is constant or uses state information only. In order for the practical implementation to strike the tradeoff between the variance reduction performance and computational complexity for maintaining the state/environment-dependent baselines, we propose a variance reduced domain randomization (VRDR) approach for policy gradient methods, which can improve the sample efficiency while maintaining a reasonable number of baselines associated with a set of specifically designed environment subspaces. Our main contributions can be summarized as follows.

- **Theoretically optimal state/environment-dependent baseline for DR.** For the policy gradient in a variety of environments that differ in dynamics, we derive theoretically an optimal baseline that depends on both the states and environments. We further quantify the variance reduction improvement achieved by the proposed state/environment-dependent baseline over two common choices of baselines for DR, i.e., the constant and state-dependent baselines.

- **Criterion for constructing practical state/subspace-dependent baseline.** Since the accurate estimation of state/environment-dependent baseline for each possible environment is infeasible during practical implementation of RL, we propose to alternatively divide the entire space of environment parameters into a limited number of subspaces, and estimate instead the optimal baseline for every pair of state and environment subspace. We further show that the clustering of environments into subspaces should follow the policy's expected returns on these environments, which can guarantee an improvement of variance reduction over the state-dependent baseline.

- **Variance reduced domain randomization (VRDR) with empirical evaluation.** To strike the tradeoff between the variance reduction performance and computational complexity for maintaining optimal baselines, we develop a variance reduced domain randomization (VRDR) approach for policy gradient methods. Specifically, VRDR learns an acceptable number of baselines for each pair of state and environment subspace, where the environment subspaces are determined based on the above criterion. We then conduct experiments on six robot control tasks with their fundamental physical parameters randomized, demonstrating that VRDR can accelerate the convergence of policy training in DR settings, as compared to the standard state-dependent baselines. In some specific tasks, VRDR can even achieve a higher reward.

## 2 BACKGROUND

**Notation.** Under the standard reinforcement learning (RL) setting, the environment is modeled as a Markov decision process (MDP) defined by a tuple $< \mathcal{S}, \mathcal{A}, \mathcal{T}_p, \Phi >$, where $\mathcal{S}$ and $\mathcal{A}$ denote the state and action spaces, respectively. For the convenience of derivation, we assume that they are finite. $\mathcal{T}_p : \mathcal{S} \times \mathcal{A} \times \mathcal{S} \to [0, 1]$ is the environment transition model that is essentially determined by environment parameter $p \in \mathcal{P}$, with $\mathcal{P}$ denoting the space of environment parameters. In robot control, for example, environment parameter $p$ can be a vector containing the rolling friction of each joint and the mass of torso and arms. Throughout the rest of this paper, by environment $p$ we mean that the environment has dynamics determined by parameter $p$. $\Phi : \mathcal{S} \times \mathcal{A} \to \mathbb{R}$ is the reward function. At each time step $t$, the agent observes its state $s_t \in \mathcal{S}$ and takes an action $a_t \in \mathcal{A}$ under the guidance of policy $\pi_\theta(a_t|s_t)$ parameterized by $\theta$. It will then receive a reward $r_t = \Phi(s_t, a_t)$, while the environment shifts to the next state $s_{t+1}$ with probability $\mathcal{T}(s_{t+1}|s_t, a_t, p)$. The goal of standard RL is to search for a policy $\pi$ that maximizes the expected discounted return $\eta(\pi, p) = \mathbb{E}_\tau [R(\tau)]$ over all possible trajectories $\tau = \{s_t, a_t, r_t, s_{t+1}\}_{t=0}^\infty$ of states and actions, where $R(\tau) = \sum_{t=0}^\infty \gamma^t r_t$ and $\gamma \in [0, 1]$ is the discount factor. We can then define the state value function as $V_\pi(s, p) = \mathbb{E}\left[\sum_{k=0}^\infty \gamma^k r_{t+k}|s_t = s\right]$, the action value function as $Q_\pi(s, a, p) = \mathbb{E}\left[\sum_{k=0}^\infty \gamma^k r_{t+k}|s_t = s, a_t = a\right]$, and the advantage function as $A_\pi(s, a, p) = Q_\pi(s, a, p) - V_\pi(s, p)$.

**Policy gradient methods for DR.** In DR, the environment parameter p is a random variable that follows the probability distribution $P$ over $\mathcal{P}$. For the convenience of derivation, we assume a finite en-

vironment parameter space $\mathcal{P}$ with cardinality $|\mathcal{P}|$. By introducing DR, the goal of policy optimization is to maximize the expected return over all possible environment parameters: $\mathbb{E}_{p \sim P}[\eta(\pi, p)]$. The policy gradient with action-independent baseline (Sutton & Barto, 2018) can be formulated as:

$$\nabla_\theta \mathbb{E}_{p \sim P}[\eta(\pi, p)] = \mathbb{E}_P\left[\mathbb{E}_{\mu_\pi^p, \pi}\left[\nabla_\theta \log \pi_\theta(a|s)\left[Q_\pi(s, a, p) - b\right]\right]\right], \qquad (1)$$

where we define $\mu_\pi^p(s) = \sum_{t=0}^{\infty} \gamma^t P_\pi(s_t = s|p)$ as the discounted state visitation frequency, with $P_\pi(s_t = s|p)$ denoting the probability of shifting from the initial state $s_0$ to state $s$ after $t$ steps under policy $\pi$ in environment $p$. For convenience, we further denote $g(\theta, s, a, p) \triangleq \nabla_\theta \log \pi_\theta(a|s)[Q_\pi(s, a, p) - b]$, which is the gradient estimator for the state-action pair under environment $p$. As long as the baseline $b$ is action-independent, we have $\mathbb{E}_a[\nabla_\theta \log \pi_\theta(a|s)b] = \nabla_\theta \mathbb{E}_a[b] = 0$ and thus $\mathbb{E}_{P, \mu_\pi^p, \pi}[g] = \mathbb{E}_{P, \mu_\pi^p, \pi}[\nabla_\theta \log \pi_\theta(a|s)Q_\pi(s, a, p)] \triangleq \mathbb{E}[g]$, where the subscript is dropped for convenience. Therefore, by subtracting this action-independent baseline $b$, the variance of gradient estimator $Var(g) = \mathbb{E}[g^\mathrm{T}g] - \mathbb{E}[g]^\mathrm{T}\mathbb{E}[g]$ can be reduced without introducing any bias (Greensmith et al., 2004).

## 3    OPTIMAL BASELINES FOR DOMAIN RANDOMIZATION

To derive the optimal baselines for DR, we formulate the following optimization problem that aims to minimize the variance of gradient estimate w.r.t. the baseline $b$ (Greensmith et al., 2004):

$$\min_b \mathbb{E}\left[\underbrace{\left[G(a, s)\left[Q_\pi(s, a, p) - b\right]\right]^2}_{g^\mathrm{T}g}\right] - \mathbb{E}[g]^\mathrm{T}\mathbb{E}[g] \Leftrightarrow \min_b \mathbb{E}_{P, \mu_\pi^p, \pi}\left[G(a, s)\left[Q_\pi(s, a, p) - b\right]^2\right] \quad (2)$$

where we denote $G(a, s) \triangleq \nabla_\theta \log \pi_\theta(a|s)^\mathrm{T}\nabla_\theta \log \pi_\theta(a|s)$. Due to its independence of $b$, the second term $\mathbb{E}[g]^\mathrm{T}\mathbb{E}[g]$ does not affect the minimizer, and is thus omitted on RHS of Eq. (2). In the following, we first derive for DR two common choices of baselines that are constant or depend on the state only. We then propose the optimal state/environment-dependent baseline, and show its ability to further reduce the variance incurred by the randominzation of environment parameters.

### 3.1    TWO COMMON CHOICES OF ACTION-INDEPENDENT BASELINES

**Optimal constant baseline.** We first consider a constant baseline $b_c$ that depends neither on the action nor on the state. The optimization problem in Eq. (2) then becomes minimizing the expectation of a quadratic function, which can be proved to be convex. Referring to the detailed derivation in Appendix A.1, the optimal constant baseline $b_c^*$ for DR-based policy gradient methods is:

$$b_c^* = \frac{\mathbb{E}_{P, \mu_\pi^p, \pi}\left[G(a, s)Q_\pi(s, a, p)\right]}{\mathbb{E}_{P, \mu_\pi^p, \pi}\left[G(a, s)\right]} = \mathbb{E}_{P, \mu_\pi^p}\left[V'(s, p)\right]. \qquad (3)$$

This optimal baseline $b_c^*$ can be understood as the expected state value function $V'(s, p)$ over all states and possible environments, where the state value function $V'(s, p)$ is computed as the weighted average of the action value function: $V'(s, p) = \mathbb{E}_\pi\left[\frac{G(a, s)}{\mathbb{E}_{P, \mu_\pi^p, \pi}[G(a, s)]}Q_\pi(s, a, p)\right]$.

**Optimal state-dependent baseline.** As in the standard RL, the utilization of state-dependent baseline in the DR setting can be considered as an expected value function $b(s) = \mathbb{E}_P\left[\mathbb{E}_\pi[Q_\pi(s, a, p)]\right]$ for each state over all possible environments, which predicts the expected return over the distribution of dynamics caused by the variation of environment parameters. Still referring to Appendix A.1, the optimal state-dependent baseline for DR is given as follow:

$$b^*(s) = \frac{\mathbb{E}_{P(p|s)}\mathbb{E}_\pi[G(a, s)[Q_\pi(s, a, p)]]}{\mathbb{E}_\pi[G(a, s)]}. \qquad (4)$$

### 3.2    OPTIMAL STATE/ENVIRONMENT-DEPENDENT BASELINE

In DR, the randomization of environment parameters imposes additional variability for the estimate of gradients, which aggravates the instability of RL training process. In the following subsections,

we show that this variance can be effectively reduced by further incorporating into the baseline the information about varying environments. To this end, we propose a state/environment-dependent baseline $b(s, \mathcal{P}) = \{b(s, p_i)\}_{i=1}^{|\mathcal{P}|}$ for DR, and rewrite the variance minimization in Eq. (2) as:

$$\min_{b(s, \mathcal{P})} \mathbb{E}_{p \sim P, \mu_\pi^p} \left[ \mathbb{E}_\pi \left[ G(a, s) \left[ Q_\pi(s, a, p) - b(s, p) \right]^2 \right] \right]. \tag{5}$$

In Appendix A.1, following the similar deduction as in (Greensmith et al., 2004), the **optimal state/environment baseline** can be derived as:

$$b^*(s, \mathcal{P}) = \{b^*(s, p_i)\}_{i=1}^{|\mathcal{P}|}, \text{ where } b^*(s, p_i) = \frac{\mathbb{E}_\pi \left[ G(a, s) Q_\pi(s, a, p_i) \right]}{\mathbb{E}_\pi \left[ G(a, s) \right]}, \tag{6}$$

which indicates requirements of maintaining a specific baseline for each environment to incorporate the environment-specific information. Note that if environment parameter $p_i$ is time-varying along episodes and considered as stochastic input process that partially affects the dynamics, this optimal baseline $b^*(s, \mathcal{P})$ is equivalent to the optimal input-driven baseline as derived by Mao et al. (2018).

### 3.3 Variance Reduction Improvement of $b^*(s, \mathcal{P})$

**Theorem 1.** *Let $Var^{b(s)}(g)$ and $Var^{b^*(s, \mathcal{P})}(g)$ denote the variance of gradient estimator by incorporating an arbitrary state-dependent baseline $b(s)$ and the optimal state/environment-dependent baseline $b^*(s, \mathcal{P})$, respectively. Compared to the state-dependent baseline $b(s)$, the variance can be further reduced by $b^*(s, \mathcal{P})$, with the following improvement:*

$$Var^{b(s)}(g) - Var^{b^*(s, \mathcal{P})}(g) = \mathbb{E}_{P, \mu_\pi^p} \left[ \sqrt{\mathbb{E}_\pi \left[ G(a, s) \right]} b(s) - \frac{\mathbb{E}_\pi \left[ G(a, s) Q_\pi(s, a, p) \right]}{\sqrt{\mathbb{E}_\pi \left[ G(a, s) \right]}} \right]^2. \tag{7}$$

*Proof.* See Appendix A.2. $\qquad\square$

Guided by Theorem 1, we can then quantify the variance improvement achieved by the optimal state/environment-dependent baseline $b^*(s, \mathcal{P})$ over the optimal state-dependent baseline $b^*(s)$ by letting $b(s) = b^*(s)$, as shown in the following corollary.

**Corollary 1.** *Let $Var^{b^*(s)}(g)$ denote the variance of gradient estimator with the optimal state-dependent baseline $b^*(s)$. The variance reduction improvement of $b^*(s, \mathcal{P})$ over $b^*(s)$ is:*

$$Var^{b^*(s)}(g) - Var^{b^*(s, \mathcal{P})}(g) = \mathbb{E}_{P, \mu_\pi^p} \left[ \frac{\left( \mathbb{E}_{P(p|s)} \left[ \mathbb{E}_\pi [G(a, s) Q_\pi(s, a, p)] \right] - \mathbb{E}_\pi \left[ G(a, s) Q_\pi(s, a, p) \right] \right)^2}{\mathbb{E}_\pi \left[ G(a, s) \right]} \right]. \tag{8}$$

Since the optimal constant baseline $b_c^*$ can be thought as letting $b(s) = b_c^*$ for each state $s \in \mathcal{S}$, we show in Appendix A.3 that further variance reduction is achieved by $b^*(s, \mathcal{P})$ over $b_c^*$. Therefore, we analytically justify that by incorporating the additional information about environments, the optimal state/environment-dependent baseline $b^*(s, \mathcal{P})$ can obtain the minimum variance for DR within the baselines that consider both state and environment parameters.

## 4 Variance Reduced Domain Randomization

For the single environment case in standard RL, the theoretically optimal state-dependent baseline is rarely used in practice due to the computational concern (Wu et al., 2018). Rather, for both implementational and conceptual benefit, a common alternative choice of $b(s)$ is the state value function $V(s)$. Similarly in DR, even though the optimal state/environment-dependent baseline is known and given in Eq. (6), the computation of $|\mathcal{P}|$ baselines for all the possible environments requires a very high computational and sampling complexity, which is therefore infeasible during the practical training process. Fortunately, similar system dynamics often emerge on environments having the similar parameters, on which the same policy will also obtain the similar expected returns.

In the following, we aim to strike a tradeoff between the variance reduction performance and the computational complexity required for maintaining multiple state/environment-dependent baselines.

Specifically, we propose to maintain an acceptable number $M$ of state/environment-dependent baselines in practice, where $M < |\mathcal{P}|$. The entire environment parameter space $\mathcal{P}$ is then divided as $M$ subspaces $\{\mathcal{P}_j\}_{j=1}^M$, each of which has a size $|\mathcal{P}_j|$. For each subspace $\mathcal{P}_j$, we can compute the optimal state-only baseline as $b^*(s, \mathcal{P}_j) = \mathbb{E}_{P_j(p|s)}\mathbb{E}_\pi[G(a|s)Q_\pi(s,a,p)]/\mathbb{E}_\pi[G(a,s)]$. We then denote the state/subspace-dependent baseline as $b^M(s, \mathcal{P}) = \{b^*(s, \mathcal{P}_j)\}_{j=1}^M$. Assuming a uniform sampling on $\mathcal{P}$, and referring to Appendix A.4, the total variance of the gradient estimator by incorporating the state/subspace-dependent baseline $b^M(s, \mathcal{P})$ can be written as:

$$Var^M(g) = \sum_{j=1} \frac{|\mathcal{P}_j|}{|\mathcal{P}|} Var(g_M^j), \ g_M^j = \nabla_\theta \log \pi_\theta(a|s)\left(Q_\pi(s,a,p_i^j) - b^*(s, \mathcal{P}_j)\right), p_i^j \in \mathcal{P}_j, \ (9)$$

where $p_i^j$ is the $i$-th environment parameter in subspace $j$ with sampling distribution $\frac{1}{|\mathcal{P}_j|}$, and $Var(g_M^j)$ is computed over $p_i^j \in \mathcal{P}_j$, following s $\sim \mu_\pi(\cdot|p_i^j)$ and a $\sim \pi$.

**Theorem 2.** *The variance difference of gradient estimate between the optimal state-dependent baseline and the state/subspace-dependent baseline is:*

$$Var^{b^*(s)}(g) - Var^M(g) \tag{10}$$

$$= \sum_{j=1}^M \sum_{i=1}^{|\mathcal{P}_j|} \frac{1}{|\mathcal{P}|} \sum_s \mu_\pi(s|p_i^j) \frac{\left[\left(\mathbb{E}_{P(p|s)}[Y_p(s)] - Y_{p_i^j}(s)\right)^2 - \left(\mathbb{E}_{P_j(p|s)}[Y_p(s)] - Y_{p_i^j}(s)\right)^2\right]}{\mathbb{E}_\pi[G(a,s)]},$$

*where $Y_p(s) \triangleq \mathbb{E}_\pi[G(a,s)Q_\pi(s,a,p)]$. For a policy $\pi$ to be updated at a certain step, and assuming the uniform distribution for sampling environment parameters, we have $Var^{b^*(s)}(g) \geq Var^M(g)$ if for each subspace $\mathcal{P}_j$:*

$$Var^{\mathcal{P}}(Y_p(s)) \geq Var^{\mathcal{P}_j}(Y_p(s)), \tag{11}$$

*where the variance $Var^{\mathcal{P}}(Y_p(s))$ and $Var^{\mathcal{P}_j}(Y_p(s))$ is computed over $\mathcal{P}$ and $\mathcal{P}_j$, respectively.*

*Proof.* See Appendix A.5. $\square$

Theorem 2 provides a practical guideline for constructing subspaces to reduce the variance of gradient estimate, by holding the variance of $Y_p(s)$ in each subspace not exceeding that in the entire environment space. However, it still introduces extra computation to estimating $G(a,s)$ in $Y_p(s)$. When the inner product of log policy $G(s,a)$ is loosely correlated to the $Q$-value function[1], we have $\mathbb{E}_a[G(a,s)Q_\pi(s,a,p)] = \mathbb{E}_a[G(a,s)]\mathbb{E}_a[Q_\pi(s,a,p)]$, and Eq. (11) is thus equivalent to:

$$Var^{\mathcal{P}}(V_\pi(s,p)) \geq Var^{\mathcal{P}_j}(V_\pi(s,p)), \tag{12}$$

It indicates that by clustering the entire environment space into some subspaces and holding the variance of expected return in each subspace not greater than that in the entire environment space, the state/subspace-dependent baseline can further reduce the variance of gradient estimate compared to the optimal state-only baseline. Note that by increasing the number of subspaces, the state/subspace-dependent baseline $b^M(s, \mathcal{P})$ will approach the optimal sate/environment-dependent baseline $b^*(s, \mathcal{P})$. However, we argue that the more baselines we employ, the more samples are required for estimating these baselines. As will be shortly shown by the experiment, increasing $M$ may not monotonically improve the policy's performance during the training process.

Based on our theory, we propose a practical variance reduced domain randomization approach (VRDR) for policy gradient and summarize it in Algorithm 1, where the state/subspace-dependent baseline $b(s, \mathcal{P}_j) = \mathbb{E}_{p \sim P_j}V(s,p) \triangleq V(s, \mathcal{P}_i)$ is employed for each state and subspace pair. At each

---

[1]Take the Gaussian policy that is commonly used in policy gradient methods for example. The policy takes $s$ as input and outputs for action selections a Gaussian distribution with the expectation $m_a$ and variance $\sigma^2$. The action $a = m_a$ has the minimum gradient of zero, i.e., $G(m_a, s) = 0$, while this state and action pair $(m_a, s)$ is not correlated with the $Q$ value before the trained policy obtains a high $Q$ value.

---

**Algorithm 1** Variance Reduced Domain Randomization (VRDR) for Policy Gradient

---

1: **Initialize** policy $\pi_0$, number of environment parameters sampled per iteration $H$, number of subspaces $M$, clustering interval $N_c$, maximal number of iterations $N$, subspace prototypes $\boldsymbol{u} = \{u_i\}_{i=1}^M$, baseline $V(s|\mathcal{P}_i)$ for each subspace $\mathcal{P}_i$, and cluster labels $\boldsymbol{l} = \{l_i\}_{i=1}^M$ as shuffled with $\{i\}_{i=1}^M$.
2: **for** $k = 0$ to $N - 1$ **do**
3:    **if** $k \% N_c == 0$ **then**
4:       Sample $H$ environments $\{p_c\}_{c=0}^{H-1}$ with $L$ trajectories for each environment $p_c$.
5:       Compute $\Omega = \{\hat{V}_\pi(p_c)\}_{c=1}^H$, where $\hat{V}_\pi(p_c) = \sum_{j=0}^{L-1} R(\tau_{c,j})/L$ is the average return of environment $p_c$.
6:       Apply clustering method on $\Omega$ to get the cluster set $\{\mathcal{C}_i\}_{i=1}^M$, with each $\mathcal{C}_i$ labeled by $i$.
7:       Determine corresponding prototypes $\boldsymbol{u}'$ by computing $u_i' = \sum_{j \in \mathcal{C}_i} p_j / |\mathcal{C}_i|$.
8:       Determine $\min_{\boldsymbol{l}} \sum_i |u_{l_i}' - u_i|$ where $\boldsymbol{l} = \{l_i\}$, update prototypes by $u_i = u_{l_i}'$, and relabel cluster $i$ with $l_i$.
9:    **else**
10:       Sample a set of environment parameters $\{p_e\}_{e=0}^{H-1}$ uniformly and determine their cluster label by $\min_i |p_e - u_i|$ to obtain the cluster set $\{\mathcal{C}_i\}_{i=1}^M$.
11:       Sample $L$ trajectories $\{\tau_{e,j}\}_{j=0}^{L-1}$ for each environment $p_e$ using policy $\pi_k$.
12:    **end if**
13:    Use PPO for policy optimization to get the updated policy $\pi_{k+1}$ on trajectories, with the baseline $V(s|\mathcal{P}_i)$ of the cluster that has been grouped to.
14:    Update $V(s|\mathcal{P}_i)$ on the trajectories in cluster $\mathcal{C}_i$.
15: **end for**

---

iteration $k$, we sample $M$ environments and group them into $M$ clusters by finding the nearest subspace prototype. Then, we train the policy on the trajectories sampled on each environment with the corresponding baseline $b(s, \mathcal{P}_j)$ and update the baseline $b(s, \mathcal{P}_j)$ on the corresponding trajectories. During the clustering phase, as guided by Eq. (12), we perform the clustering method on $\{p_c\}_{c=1}^{H-1}$ w.r.t. the average return $\hat{V}_\pi(p_c)$ to get the $M$ clusters with corresponding subspace prototypes $u_i'$. The clustering methods are usually applied to group a fixed dataset, with the labeling done once and for all. In VRDR, however, we perform clustering multiple times during training, hence the prototypes between two clustering phases would be similar but have different labels. To avoid the case where the new prototype $u_i'$ close to a certain current prototype is assigned with a different label, we relabel these new prototypes $\boldsymbol{u}'$, and update the current ones $\boldsymbol{u}$ according to $\boldsymbol{u}'$ as described in Line 8 of Algorithm 1, where we aim to find a set of labels $\boldsymbol{l} = \{l_i\}$ that minimizes the distance $|u_{l_i}' - u_i|$ between new prototype $u'$ and current prototype $u$. Then, after the clustering phase, the subspace $\mathcal{P}_j$ can be reconstructed according to the sampled environments in cluster $\mathcal{C}_j$. We utilize loss function $\mathcal{L}_V = \sum_{s_t \in \boldsymbol{\tau}_i} |V(s_t, \mathcal{P}_i) - \sum_{t'=t}^{\infty} \gamma^{t'-t} r_t|^2$ to update the baseline $V(s, \mathcal{P}_i)$ for each cluster $\mathcal{C}_i$.

It has been proved by Ghadimi & Lan (2013) that the optimization of gradient variance can accelerate the convergence of gradient descent methods. In Appendix A.7, we show that our proposed VRDR method gains a $\kappa = (\kappa > 1)$ acceleration on the convergence of policy gradient methods, as compared to the direct application of optimal state-dependent baseline for DR.

## 5 EXPERIMENTS

In this section, we evaluate our proposed VRDR method on six simulated robot control tasks, where the fundamental environment parameters that affect dynamics are randomized for the generalization of trained policy. The experimental environments are implemented based on the robot control simulator as described in (Brockman et al., 2016). We compare VRDR with two baseline algorithms: the uniform domain randomization (DR) (Mehta et al., 2020) and the multiple value network (MVN) (Mao et al., 2018). We utilize the proximal policy optimization (PPO) (Schulman et al., 2017) to train a generalized policy for all the comparison algorithms. In DR, we uniformly sample the environment parameter $p$ and collect trajectories by using the current policy on the sampled environments for policy update, where a value function that is learned from trajectories from all sampled environments is employed as the state-dependent baseline. In MVN, multiple value functions are learned for the pre-sampled environments at the beginning of training, while each value function $V_i$

Table 1: Range of parameters for each environment.

| Environment | Environment Parameters | Training Range |
|---|---|---|
| Walker2D/Hopper/Halfcheetah | Material Density | $[750, 1250]$ |
| | Sliding Friction | $[0.5, 1.1]$ |
| Pendulum | Pole length | $[0.5, 1.5]$ |
| Pendulum2D | Pole mass | $[0.5, 1.5]$ |
| | Pole length | $[0.5, 1.5]$ |
| InvertedDoublePendulum | Pole1 length | $[0.50, 1.00]$ |
| | Pole2 length | $[0.50, 1.00]$ |

is utilized as a baseline for the gradient estimator of trajectories collected on environment parameters that are closest to the corresponding pre-sampled $p_i$. For VRDR, we apply hierarchical cluster method (Jain, 2010) to divide the entire environment space during training, and group the trajectories collected at each interaction phase as described in Algorithm 1. See Appendix A.14 for detail of hierarchical clustering, and Appendix A.10 for comparison of clustering methods. We keep all the hyper-parameters about policy optimization for PPO the same as the implementation in Brockman et al. (2016), and purely discuss the gain brought by variance reduction for gradient estimate in policy gradient methods. Our experiments are designed to answer the following questions:

- Can VRDR effectively improve the convergence rate of the policy training. Will the final reward achieved by the converged policy trained by VRDR outperform the other two baseline algorithms?

- How would the policy performance of all the three algorithms degrade on the out-of-distribution unseen environments/dynamics?

- How would the specific hyper-parameters in VRDR affect the performance of training process.

## 5.1 TRAINING CURVES WITH RANDOMIZED DYNAMICS

The six robot control tasks are as follows. 1) **Walker2D**: control a planar biped robot to run as fast as possible; 2) **Hopper**: control a planar monopod robot to hop as fast as possible; 3) **Halfcheetah**: control a planar cheetah robot to run fast; 4 & 5) **Pendulum & Pendulum2D**: apply the force to swing a pendulum and keep it upright; 6) **InvertedDoublePendulum**: control a cart (attached to a two-link pendulum) to balance the whole system and keep the pendulum upright. We modify the environment parameters in the configure file to generate six environments with the same control task and different dynamics. The possible sampling range of the environment parameters is shown in Table 1. Specifically, we randomize the material density that determines the mass and inertia, and the sliding friction acting along the tangent plane in Walker2D, Hopper, and Halfcheetah. We randomize the pole length, respectively, in Pendulum, Pendulum2D and InvertedDoublePendulum, and the pole mass additionally in Pendulum2D.

We run the training process on environments sampled on the preset parameter range for the same number of iterations $N$ until all the algorithms are converged. At each iteration $k$, we generate $H = 100$ environments by uniformly sampling the parameter range in all algorithms. We run at least 15 seeds for each algorithm on all the environments to obtain the training results. The policy and value networks in each algorithm are trained for the same number of epochs and sampled trajectories at each iteration. We show the training curves of the six tasks, respectively, in Figs. 1(a)-1(f). The solid curve is the average returns on all the seeds at a certain iteration, while the shaded area denotes the standard deviations. It can be seen that by applying DR on PPO, we can improve the average return on the whole training range, while VRDR can accelerate the training process on the six tasks. In some specific tasks, such as Walker2D, Hopper and Halfcheetah, VRDR even achieves a higher average return than DR and MVN. In balancing tasks like Pendulum and Pendulum2D, VRDR has shown a significant acceleration on the convergence of policy training.

## 5.2 GENERALIZATION TO UNSEEN ENVIRONMENTS

In this subsection, we evaluate the generalization capability of trained policies on the unseen range during training. We learn 15 policies for DR, MVN and VRDR, respectively, on training environments with different random seeds and apply the trained policies on corresponding testing unseen environments. The generalization performance for each algorithm is measured by the average score over 15 policies. The generalization evaluation of DR, MVN and VRDR are shown in Figs. 2(a) and 2(b). VRDR shows the best generalization performance on Walker2D. DR has a broader range of generalization than MVN, although MVN achieves higher score on the training range than DR.

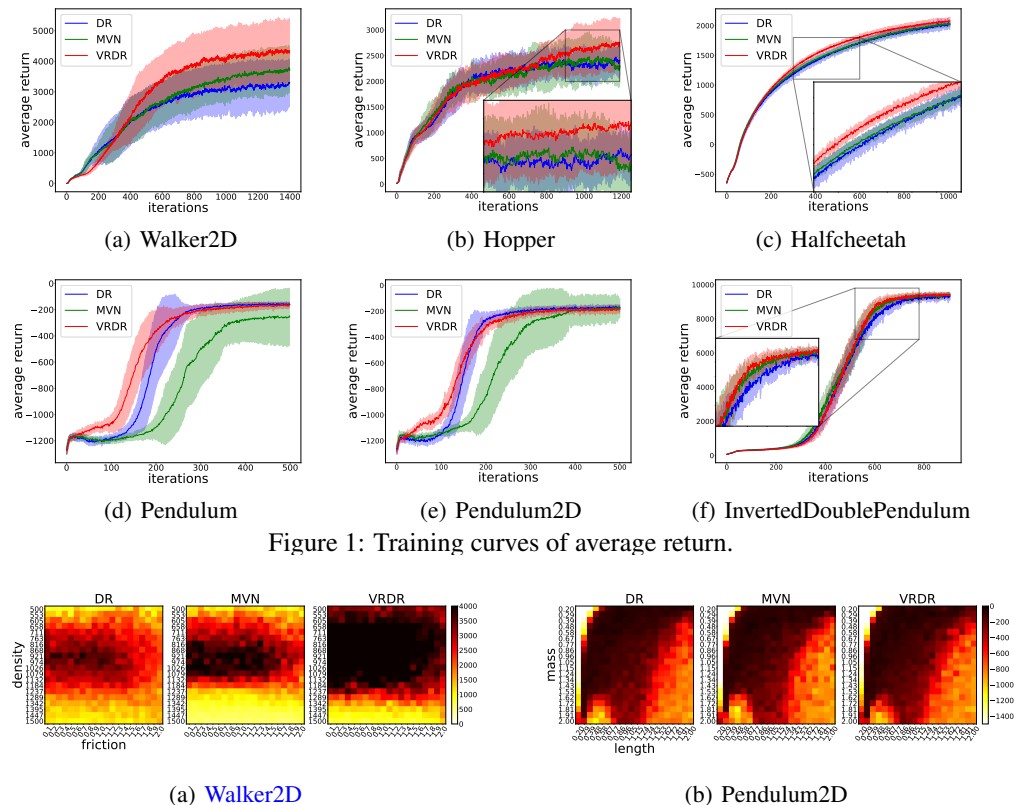

Figure 1: Training curves of average return.

(a) Walker2D

(b) Pendulum2D

Figure 2: Heatmap of return in unseen environments on Walker2D and Pendulum2D with policies trained by DR, MVN and VRDR in the training environments.

For Pendulum2D, however, all the algorithms present a close performance. This result indicates that the generalization performance is consistent with the average return achieved after convergence in training, and would not be affected by the convergence rate.

### 5.3 ABLATION STUDY OF VRDR

**Number of baselines** $M$**:** In Theorem 1, we show that by applying $b^*(s, \mathcal{P})$, we maintain a state/environment baseline $b^*(s, p_i)$ for each environment $p_i$ and can obtain the minimum variance of gradient estimate. Theoretically, increasing $M$ to $|\mathcal{P}|$ will achieve a better variance reduction improvement. Figs. 3(a)-3(c) show the training curves of VRDR on Walker2D, Pendulum, and Pendulum2D with different numbers of baselines. Note that VRDR is reduced to DR when setting $M = 1$. We find that the increase of $M$ will decrease the number of training samples for approximation of each baseline if we sample the same number of environment at each iteration, which hence retards the convergence rate of VRDR. This suggests that we can set $M$ to a small number in practice to strike the tradeoff between the variance reduction and computational complexity.

**Clustering interval** $N_c$**:** Each new clustering result may change the group that each environment belong to, hence altering the value function and affecting the policy training in the next clustering interval. The underlying dynamic mechanisms vary in different environments, and thus the shifting of value function may affect the training with varying degrees. We show the training curves of VRDR with different clustering interval $N_c$ on Walker2D, Hopper, and Halfcheetah in Figs. 4(a)-4(c), and the clustering results in Appendix A.9. The results suggest the need of a specific tuning of $N_c$ for each task.

### 6 RELATED WORK

State-of-the-art methods that have endeavored to tackle the generalization and robustness of DRL, including DR, meta RL and robust RL, all implicitly or explicitly impose diversity on the policy training. DR and meta RL both tend to learn a policy that can generalize to a variety of environments (Jiang et al., 2021; Lin et al., 2020), which may differ in dynamics and reward function during training. By introducing an adversary that acts to disturb the dynamics or the observation of the environment during training (Zhang et al., 2020), the learned policy can gain robustness on deployment.

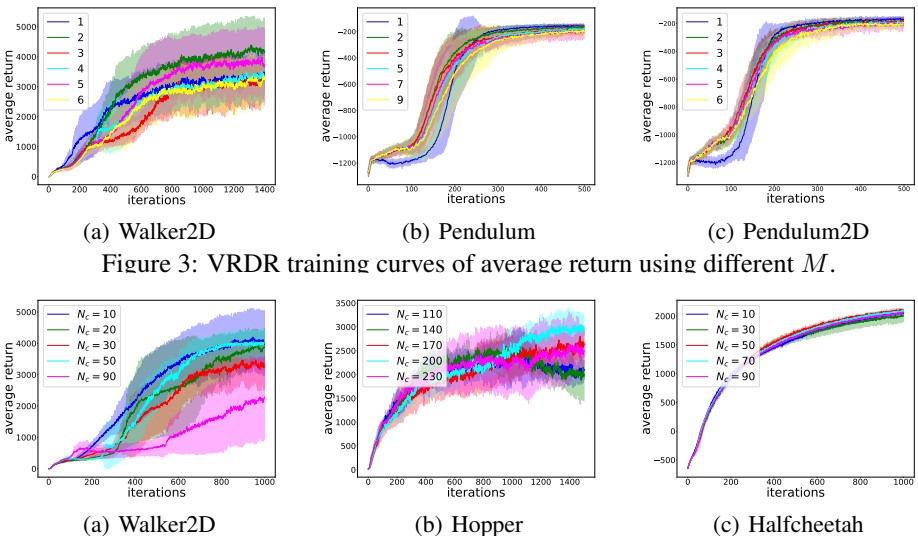

Figure 3: VRDR training curves of average return using different $M$.

Figure 4: VRDR training curves of average return using different $N_c$.

The introduced diversity for training environments enables the generalization of DRL policy, while the consequent high variance aggravates the sample efficiency of DRL training.

For policy optimization under DR, Mehta et al. (2020) declare that the uniform sampling of DR may lead to suboptimal and high-variance policy, and propose active DR to encourage the sampling distribution to select informative environments. Several works have been proposed to optimize the sampling distribution for DR (Andrychowicz et al., 2020; Paul et al., 2019), while the baseline optimization is seldom considered in the policy search using policy gradient and the direct application of state-dependent baseline is commonly used as formulated in Eq. (4).

Policy gradient methods usually suffer high variance due to the estimation of gradient from sampled trajectories. By learning a $Q$-function (critic) instead of using its sampling estimation, actor-critic methods reduce the variance with the expense of introducing a bias (Sutton & Barto, 2018). Another line of research follows the idea of control variate (Cheng et al., 2020), by subtracting an commonly used state-dependent baseline from the gradient estimate to reduce the variance without introducing bias. By assuming the independence of each dimension in the action vector, Wu et al. (2018) propose a action/state-dependent baseline, which reduces more variance than the state-only baseline. These methods are designed for standard RL and require further discussion under the context of DR. Variance reduced gradient algorithms like stochastic variance-reduced gradient (SVRG) (Johnson & Zhang, 2013) and momentum-based variance reduction (Cutkosky & Orabona, 2019) methods have been studied in the policy gradient for RL (Huang et al., 2020; Xu et al., 2020). These methods, however, do not consider the variance reduction for the DR setting. Mao et al. (2018) propose an input-dependent baseline to reduce the variance of stochastic input process, which is similar to our state/environment-dependent baseline. Then, the multi-value-network (MVN) approach and meta-learning approach are proposed to learn the baseline. Meanwhile, Liu et al. (2019) propose to learn the per-task control variate and the meta control variate for each meta-RL task, where the per-task variate is constructed by the action-dependent baseline. In essence, the MVN and per-task control variates approaches are the same, since they both aim to learn a baseline for each input or task. And both of these meta-learning approaches are built upon model-agnostic meta learning (MAML) (Finn et al., 2017) to scale when the number of input instantiations or tasks are large.

## 7 CONCLUSION

In this paper, we have focused on reducing the high variance of gradient estimator in policy gradient methods for DR with additional randomness on environment parameters. Specifically, we derived the optimal state/environment-dependent baseline, and verified that by incorporating the environment information further variance reduction could be achieved over the optimal constant or state-only baselines. We then proposed a variance reduced domain randomization (VRDR) approach for policy gradient methods, to strike a tradeoff between the variance reduction and computational complexity in practical implementation. We have validated VRDR on several robot control tasks, demonstrating an overall faster convergence speed of policy training, and even better policy performance in some specific tasks.

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

# A  APPENDIX

## A.1  DERIVATION OF OPTIMAL BASELINES

Following the derivation in (Greensmith et al., 2004), we now give the derivation of three kinds of optimal baselines by solving the optimization problem as proposed in Eq. (2).

**Derivation of optimal constant baseline.**

It can be verified that the optimization problem in Eq. (2) is convex w.r.t. the constant baseline $b_c$, and we denote:

$$\min_{b_c} F(b_c) \triangleq \mathbb{E}_{P,\mu_\pi^p,\pi} \left[ G(a,s) \left[ Q_\pi(s,a,p) - b_c \right]^2 \right]. \tag{13}$$

By letting the first-order derivative equal to zero:

$$\frac{dF(b_c)}{db_c} = -2\mathbb{E}_{P,\mu_\pi^p,\pi} \left[ G(a,s)[Q_\pi(s,a,p) - b_c] \right] = 0, \tag{14}$$

we have the optimal constant baseline for DR as:

$$b_c^* = \frac{\mathbb{E}_{P,\mu_\pi^p,\pi}[G(a,s)Q_\pi(s,a,p)]}{\mathbb{E}_{P,\mu_\pi^p,\pi}[G(a,s)]}. \tag{15}$$

**Derivation of optimal state-dependent baseline.**

The variance minimization problem w.r.t. $b(s)$ is denoted as:

$$\min_{b(s)} F(b(s)) \triangleq \mathbb{E}_{P,\mu_\pi^p,\pi} \left[ G(a,s)[Q_\pi(s,a,p) - b(s)]^2 \right]. \tag{16}$$

Following the generating process of joint distribution of $p$, $s$ and $a$, we have $P(p,s,a) = P(p)P(s|p)P(a|s,p) = P(p)\mu_\pi(s|p)\pi(a|s)$, where the last equality is because that policy $\pi$ is constructed without knowing $p$. The RHS of Eq. (16) can be expanded as:

$$\mathbb{E}_{p \sim P, \mu_\pi^p} \left[ \mathbb{E}_\pi \left[ G(a,s)[Q_\pi(s,a,p) - b(s,p)]^2 \right] \right]$$

$$= \sum_{i=1}^{|\mathcal{P}|} P(p_i) \sum_{s_j} \mu_\pi(s_j|p_i) \mathbb{E}_\pi \left[ G(a,s_j)[Q_\pi(s_j,a,p_i) - b(s_j,p_i)]^2 \right]$$

$$= \sum_{j=1}^{|\mathcal{S}|} \sum_{i=1}^{|\mathcal{P}|} P(p_i)\mu_\pi(s_j|p_i) \mathbb{E}_\pi \left[ G(a,s_j)[Q_\pi(s_j,a,p_i) - b(s_j,p_i)]^2 \right]. \tag{17}$$

By letting the first-order partial derivative equal to zero:

$$\frac{\partial F(b(s))}{\partial b(s)} = -2 \sum_{j=1}^{|\mathcal{S}|} \sum_{i=1}^{|\mathcal{P}|} P(p_i)\mu_\pi(s_j|p_i) \mathbb{E}_\pi \left[ G(a,s_j)[Q_\pi(s_j,a,p_i) - b(s_j)] \right] = 0, \tag{18}$$

where $b(s) = \{b(s_j)\}$, we have:

$$\sum_{i=1}^{|\mathcal{P}|} P(p_i)\mu_\pi(s_j|p_i) \mathbb{E}_\pi \left[ G(a,s_j)[Q_\pi(s_j,a,p_i) - b(s_j)] \right] = 0, \quad \text{for each } b(s_j). \tag{19}$$

It can be further written as

$$\sum_{i=1}^{|\mathcal{P}|} P(p_i,s_j) \mathbb{E}_\pi \left[ G(a,s_j)[Q_\pi(s_j,a,p_i)] \right] = \sum_{i=1}^{|\mathcal{P}|} P(p_i,s_j) \mathbb{E}_\pi \left[ G(a,s_j)[b(s_j)] \right], \tag{20}$$

and

$$\sum_{i=1}^{|\mathcal{P}|} P(s_j)P(p_i|s_j) \mathbb{E}_\pi \left[ G(a,s_j)[Q_\pi(s_j,a,p_i)] \right] = \sum_{i=1}^{|\mathcal{P}|} P(s_j)P(p_i|s_j) \mathbb{E}_\pi \left[ G(a,s_j) \right] b(s_j), \tag{21}$$

where $P(s_j)$ can be further canceled on both sides. We can then get the optimal state-dependent baseline for $s_j$ as

$$b^*(s_j) = \frac{\sum_{i=1}^{|\mathcal{P}|} P(p_i|s_j)\mathbb{E}_\pi\Big[G(a,s_j)[Q_\pi(s_j,a,p_i)]\Big]}{\sum_{i=1}^{|\mathcal{P}|} P(p_i|s_j)\mathbb{E}_\pi\Big[G(a,s_j)\Big]} = \frac{\mathbb{E}_{P(p|s_j)}\mathbb{E}_\pi\Big[G(a,s_j)[Q_\pi(s_j,a,p)]\Big]}{\mathbb{E}_\pi\Big[G(a,s_j)\Big]}. \tag{22}$$

For a continuous state space, we have

$$b^*(s) = \frac{\mathbb{E}_{P(p|s)}\mathbb{E}_\pi\Big[G(a,s)[Q_\pi(s,a,p)]\Big]}{\mathbb{E}_\pi\Big[G(a,s)\Big]}, \tag{23}$$

**Derivation of optimal state/environment-dependent baseline.**

The variance minimization problem w.r.t. $b(s,\mathcal{P})$ in Eq. (5) is denoted as:

$$\min_{b(s,\mathcal{P})} F(b(s,\mathcal{P})) = \mathbb{E}_{p\sim P, \mu_\pi^p}\Big[\mathbb{E}_\pi\Big[G(a,s)[Q_\pi(s,a,p) - b(s,p)]^2\Big]\Big]. \tag{24}$$

By letting the first-order partial dirivative equal to zero:

$$\frac{\partial F(b(s,\mathcal{P}))}{\partial b(s,p_i)} = -2\mathbb{E}_\pi\Big[G(a,s)\Big[Q_\pi(s,a,p) - b(s,p_i)\Big]\Big] = 0, \tag{25}$$

we obtain the optimal state/environment-dependent baseline $b^*(s,\mathcal{P}) = \{b^*(s,p_i)\}_{i=1}^{|\mathcal{P}|}$, where $b^*(s,p_i) = \mathbb{E}_\pi[G(a,s)Q(s,a,p_i)]/\mathbb{E}_\pi[G(a,s)]$.

## A.2 PROOF OF THEOREM 1

The variance of policy gradient using the optimal state/environment-dependent baseline $b^*(s,\mathcal{P})$ is:

$$Var^{b^*(s,\mathcal{P})}(g) = \mathbb{E}\left[G(a,s)\left[Q_\pi(s,a,p) - \frac{\mathbb{E}_\pi[G(a,s)Q(s,a,p)]}{\mathbb{E}_\pi[G(a,s)]}\right]^2\right] - \mathbb{E}[g]^{\mathrm{T}}\mathbb{E}[g]$$

$$= \mathbb{E}\left[G(a,s)\left[Q_\pi^2(s,a|p) - 2Q_\pi(s,a,p)\frac{Y_p(s)}{Z} + \frac{Y_p^2(s)}{Z^2}\right]\right] - \mathbb{E}[g^{\mathrm{T}}]\mathbb{E}[g]$$

$$= \mathbb{E}_{P,\mu_\pi^p}[X_p(s)] - \mathbb{E}_{P,\mu_\pi}\left[\frac{Y_p^2(s)}{Z}\right] - \mathbb{E}[g^{\mathrm{T}}]\mathbb{E}[g], \tag{26}$$

where we denote $X_p(s) \triangleq \mathbb{E}_\pi\big[G(a,s)Q_\pi^2(s,a,p)\big]$, $Y_p(s) \triangleq \mathbb{E}_\pi\big[G(a,s)Q_\pi(s,a,p)\big]$ and $Z \triangleq \mathbb{E}_\pi[G(a,s)]$. For an arbitrary state-dependent baseline $b(s)$, the variance difference between $b(s)$ and $b^*(s,\mathcal{P})$ is:

$$Var^{b(s)}(g) - Var^{b^*(s,\mathcal{P})}(g)$$

$$= \mathbb{E}_{P,\mu_\pi^p}\mathbb{E}_\pi\left[G(a,s)\left[(Q_\pi(s,a,p) - b(s))^2 - \left(Q_\pi(s,a,p) - \frac{Y_p(s)}{Z}\right)^2\right]\right]$$

$$= \mathbb{E}_{P,\mu_\pi^p}\mathbb{E}_\pi\left[G(a,s)\left[b^2(s) - 2Q_\pi(s,a,p)b(s) + 2Q_\pi(s,a,p)\frac{Y_p(s)}{Z} - \frac{Y_p^2(s)}{Z^2}\right]\right]$$

$$= \mathbb{E}_{P,\mu_\pi^p}\left[\mathbb{E}_\pi[G(a,s)]b^2(s) - 2\mathbb{E}_\pi[G(a,s)Q(s,a,p)]b(s) + \frac{\mathbb{E}_\pi^2[G(a,s)Q(s,a,p)]}{\mathbb{E}_\pi[G(a,s)]}\right]$$

$$= \mathbb{E}_{P,\mu_\pi^p}\left[\sqrt{\mathbb{E}_\pi[G(a,s)]}b(s) - \frac{\mathbb{E}_\pi[G(a,s)Q(s,a,p)]}{\sqrt{\mathbb{E}_\pi[G(a,s)]}}\right]^2. \tag{27}$$

### A.3 Variance Reduction Improvement of $b^*(s, \mathcal{P})$ over $b_c^*$

The application of optimal constant baseline $b_c^*$ can be considered as letting $b(s) = b_c^*$ for each state $s \in \mathcal{S}$. Hence, by direct applying Theorem 1, we can obtain the variance reduction improvement of $b^*(s, \mathcal{P})$ over $b_c^*$ as:

$$
\begin{aligned}
& Var^{b_c^*}(g) - Var^{b^*(s,\mathcal{P})}(g) \\
=& \mathbb{E}_{P,\mu_\pi^p} \left[ \sqrt{\mathbb{E}_\pi \big[ G(a,s) \big]} b_c^* - \frac{\mathbb{E}_\pi \big[ G(a,s) Q_\pi(s,a,p) \big]}{\sqrt{\mathbb{E}_\pi \big[ G(a,s) \big]}} \right]^2 \\
=& \mathbb{E}_{P,\mu_\pi^p} \left[ \mathbb{E}_\pi \big[ G(a,s) \big] \left( \frac{\mathbb{E}_{P,\mu_\pi^p,\pi} \big[ G(a,s) Q_\pi(s,a,p) \big]}{\mathbb{E}_{P,\mu_\pi^p,\pi} \big[ G(a,s) \big]} - \frac{\mathbb{E}_\pi \big[ G(a,s) Q_\pi(s,a,p) \big]}{\mathbb{E}_\pi \big[ G(a,s) \big]} \right)^2 \right].
\end{aligned}
\tag{28}
$$

### A.4 Variance of Policy Gradient Estimator by Applying $b^M(s, \mathcal{P})$

Under uniform domain randomization, we demonstrate that the variance of policy gradient in the whole environment space can be formulated as weighted sum of the variance in subspaces:

$$
\begin{aligned}
Var^M(g) &= \sum_{k=1}^{|\mathcal{P}|} \frac{1}{|\mathcal{P}|} \sum_s \mu_\pi(s|p_k) \sum_a \pi(a|s)(g(\theta,s,a,p_k) - \mathbb{E}g(\theta,s,a,p_k))^\mathrm{T}(g(\theta,s,a,p_k) - \mathbb{E}g(\theta,s,a,p_k)) \\
&= \sum_{j=1}^{M} \frac{|\mathcal{P}_j|}{|\mathcal{P}|} \sum_{i=1}^{|\mathcal{P}_j|} \frac{1}{|\mathcal{P}_j|} \sum_s \mu_\pi(s|p_i^j) \sum_a \pi(a|s)(g(\theta,s,a,p_i^j) - \mathbb{E}g(\theta,s,a,p_i^j))^\mathrm{T}(g(\theta,s,a,p_i^j) - \mathbb{E}g(\theta,s,a,p_i^j)) \\
&= \sum_{j=1}^{M} \frac{|\mathcal{P}_j|}{|\mathcal{P}|} Var(g^j),
\end{aligned}
\tag{29}
$$

where $g^j$ is the gradient estimator of environment in subspace $j$ and $p_i^j$ is the $i-$th environment in subspace $j$.

### A.5 Proof of Theorem 2

In this subsection, we study how to divide subspace to reduce variance of gradient estimator. First, we derive the variance difference in Eq. (10). For a certain state-dependent baseline $b(s)$ and an arbitrary sampling distribution $P$ of environment parameters, the variance of gradient estimate is:

$$
\begin{aligned}
Var^{b(s)}(g) &= \sum_{j=1}^{M} \sum_{i=1}^{|\mathcal{P}_j|} P(p_i) \sum_s \mu_\pi(s|p_i^j) \sum_a \pi(a|s) G(a|s) \left[ Q_\pi^2(s,a,p_i^j) - 2Q_\pi(s,a,p_i^j)b(s) + b^2(s) \right] \\
&= \sum_{j=1}^{M} \sum_{i=1}^{|\mathcal{P}_j|} P(p_i) \sum_s \mu_\pi(s|p_i^j) \left[ X_{p_i^j} - \frac{Y_{p_i^j}^2(s)}{Z} + \frac{Y_{p_i^j}^2(s)}{Z} - 2b(s)Y_{p_i^j}(s) + b^2(s)Z \right] \\
&= \sum_{j=1}^{M} \sum_{i=1}^{|\mathcal{P}_j|} P(p_i) \sum_s \mu_\pi(s|p_i^j) \left[ X_{p_i^j} - \frac{Y_{p_i^j}^2(s)}{Z} + \left( \frac{Y_{p_i^j}(s)}{\sqrt{Z}} - b(s)\sqrt{Z} \right)^2 \right].
\end{aligned}
\tag{30}
$$

The variance of gradient estimate by incorporating the state/subspace-dependent baseline is

$$
Var^M(g) = \sum_{j=1}^{M} \sum_{i=1}^{|\mathcal{P}_j|} P(p_i) \sum_s \mu_\pi(s|p_i^j) \left[ X_{p_i^j} - \frac{Y_{p_i^j}^2(s)}{Z} + \left( \frac{\mathbb{E}_{P_j(p|s)} Y_p(s)}{\sqrt{Z}} - \frac{Y_{p_i^j}(s)}{\sqrt{Z}} \right)^2 \right],
\tag{31}
$$

and variance of gradient estimate by incorporating the optimal state-dependent baseline is

$$Var^{b^*(s)}(g) = \sum_{j=1}^{M}\sum_{i=1}^{|\mathcal{P}_j|} P(p_i) \sum_s \mu_\pi(s|p_i^j) \left[ X_{p_i^j} - \frac{Y_{p_i^j}^2(s)}{Z} + \left( \frac{\mathbb{E}_{P(p|s)}Y_p(s)}{\sqrt{Z}} - \frac{Y_{p_i^j}(s)}{\sqrt{Z}} \right)^2 \right].$$

(32)

The difference between them is

$$Var^{b^*(s)}(g) - Var^M(g)$$

$$= \sum_{j=1}^{M}\sum_{i=1}^{|\mathcal{P}_j|} P(p_i) \sum_s \mu_\pi(s|p_i^j) \frac{\left[ \left( \mathbb{E}_{P(p|s)}[Y_p(s)] - Y_{p_i^j}(s) \right)^2 - \left( \mathbb{E}_{P_j(p|s)}[Y_p(s)] - Y_{p_i^j}(s) \right)^2 \right]}{\mathbb{E}_\pi[G(a,s)]}.$$

(33)

Thus, Eq. (10) can be obtained by letting $P(p_i) = \frac{1}{|\mathcal{P}|}$. Then, we discuss the dividing process of subspaces to make the variance reduction improvement $Var^{b^*(s)}(g) \geq Var^M(g)$ hold. Under the uniform sampling distribution $P(p) = \frac{1}{|\mathcal{P}|}$, we have

$$\sum_{j=1}^{M} \frac{|\mathcal{P}_j|}{|\mathcal{P}|} Var^{\mathcal{P}}(Y_p(s)) = Var^{\mathcal{P}}(Y_p(s)) = \sum_{i=1}^{|\mathcal{P}|} P(p_i|s)(\mathbb{E}_{P(p|s)}[Y_p(s)] - Y_{p_i}(s))^2$$

$$= \sum_{j=1}^{M}\sum_{i=1}^{|\mathcal{P}_j|} P(p_i^j|s)(\mathbb{E}_{P(p|s)}[Y_p(s)] - Y_{p_i^j}(s))^2$$

$$= \sum_{j=1}^{M}\sum_{i=1}^{|\mathcal{P}_j|} \frac{1}{|\mathcal{P}|} \frac{P(s|p_i^j)}{P(s)}(\mathbb{E}_{P(p|s)}[Y_p(s)] - Y_{p_i^j}(s))^2. \quad (34)$$

The first equality holds since $\sum_{j=1}^{M} \frac{|\mathcal{P}_j|}{|\mathcal{P}|} = 1$. And

$$\sum_{j=1}^{M} \frac{|\mathcal{P}_j|}{|\mathcal{P}|} Var^{\mathcal{P}_j}(Y_p(s)) = \sum_{j=1}^{M} \frac{|\mathcal{P}_j|}{|\mathcal{P}|} \sum_{i=1}^{|\mathcal{P}_j|} P_j(p_i|s)(\mathbb{E}_{P_j(p|s)}[Y_p(s)] - Y_{p_i^j}(s))^2$$

$$= \sum_{j=1}^{M} \frac{|\mathcal{P}_j|}{|\mathcal{P}|} \sum_{i=1}^{|\mathcal{P}_j|} \frac{1}{|\mathcal{P}_j|} \frac{P(s|p_i^j)}{P(s)}(\mathbb{E}_{P_j(p|s)}[Y_p(s)] - Y_{p_i^j}(s))^2$$

$$= \sum_{j=1}^{M}\sum_{i=1}^{|\mathcal{P}_j|} \frac{1}{|\mathcal{P}|} \frac{P(s|p_i^j)}{P(s)}(\mathbb{E}_{P_j(p|s)}[Y_p(s)] - Y_{p_i^j}(s))^2. \quad (35)$$

Note that $P(s|p_i^j) = \mu_\pi(s|p_i^j)$. Thus, as long as $Var^{\mathcal{P}}(Y_p(s)) \geq Var^{\mathcal{P}_j}(Y_p(s))$, we have $Var^{\mathcal{P}}(Y_p(s)) = \sum_{j=1}^{M} \frac{|\mathcal{P}_j|}{|\mathcal{P}|} Var^{\mathcal{P}}(Y_p(s)) \geq \sum_{j=1}^{M} \frac{|\mathcal{P}_j|}{|\mathcal{P}|} Var^{\mathcal{P}_j}(Y_p(s))$. Then, combing Eq. (10), we have $Var^{b^*(s)}(g) \geq Var^M(g)$.

## A.6 DOMAIN RANDOMIZATION WITH AN ARBITRARY SAMPLING DISTRIBUTION

In this subsection, we study how to divide subspace to reduce variance of gradient estimator in domain randomization with an arbitrary sampling distribution. Specifically, we prove that for an arbitrary sampling distribution of environment parameters, as long as Eq. (11) holds, we still have $Var^{b^*(s)}(g) \geq Var^M(g)$. Under arbitrary sampling distribution $P$, the variance difference of gradient estimate between the optimal state-dependent baseline and the state/subspace-dependent baseline

is:

$$Var^{b^*(s)}(g) - Var^M(g)$$

$$= \sum_{j=1}^{M} \sum_{p_i \in \mathcal{P}_j} P(p_i) \sum_{s} \mu_\pi(s|p_i) \frac{\left[ \left( \mathbb{E}_{P(p|s)}[Y_p(s)] - Y_{p_i}(s) \right)^2 - \left( \mathbb{E}_{P_j(p|s)}[Y_p(s)] - Y_{p_i}(s) \right)^2 \right]}{\mathbb{E}_\pi \left[ G(a,s) \right]},$$

(36)

where $P(p_i)$ is the sampling probability of environment $p_i$ from the entire environment space $\mathcal{P}$. Hence, $\frac{P(p_i)}{\sum_{p_i \in \mathcal{P}_j} P(p_i)}$ can be used to denote the sampling probability of environment $p_i$ within the subspace $\mathcal{P}_j$. For $Var^{\mathcal{P}_j}(Y_p(s))$, we have

$$Var^{\mathcal{P}_j}(Y_p(s)) = \sum_{p_i \in \mathcal{P}_j} \frac{P(p_i)}{\sum_{p_i \in \mathcal{P}_j} P(p_i)} \frac{P(s|p_i)}{P(s)} (\mathbb{E}_{P_j(p|s)}[Y_p(s)] - Y_{p_i}(s))^2,$$

(37)

and thus

$$\sum_{j=1}^{M} \left[ \sum_{p_i \in \mathcal{P}_j} P(p_i) \right] Var^{\mathcal{P}_j}(Y_p(s)) = \sum_{j=1}^{M} \sum_{p_i \in \mathcal{P}_j} P(p_i) \frac{P(s|p_i)}{P(s)} (\mathbb{E}_{P_j(p|s)}[Y_p(s)] - Y_{p_i}(s))^2.$$

(38)

For $Var^{\mathcal{P}}(Y_p(s))$, we have

$$\sum_{j=1}^{M} \left[ \sum_{p_i \in \mathcal{P}_j} P(p_i) \right] Var^{\mathcal{P}}(Y_p(s)) = Var^{\mathcal{P}}(Y_p(s))$$

$$= \sum_{p_i \in \mathcal{P}} P(p_i|s)(\mathbb{E}_{P(p|s)}[Y_p(s)] - Y_{p_i}(s))^2$$

$$= \sum_{j=1}^{M} \sum_{p_i \in \mathcal{P}_j} P(p_i|s)(\mathbb{E}_{P(p|s)}[Y_p(s)] - Y_{p_i}(s))^2$$

$$= \sum_{j=1}^{M} \sum_{p_i \in \mathcal{P}_j} P(p_i) \frac{P(s|p_i)}{P(s)} (\mathbb{E}_{P(p|s)}[Y_p(s)] - Y_{p_i}(s))^2, \quad (39)$$

where the first equality follows that $\sum_{j=1}^{M} \left[ \sum_{p_i \in \mathcal{P}_j} P(p_i) \right] = 1$ and the second quality is from the definition of $Var^{\mathcal{P}}(Y_p(s))$.

Note that $P(s|p_i^j) = \mu_\pi(s|p_i^j)$. Thus, as long as $Var^{\mathcal{P}}(Y_p(s)) \geq Var^{\mathcal{P}_j}(Y_p(s))$, we have $\sum_{j=1}^{M} \left[ \sum_{p_i \in \mathcal{P}_j} P(p_i) \right] Var^{\mathcal{P}}(Y_p(s)) \geq \sum_{j=1}^{M} \left[ \sum_{p_i \in \mathcal{P}_j} P(p_i) \right] Var^{\mathcal{P}_j}(Y_p(s))$. Then, combing Eq. (36), we have $Var^{b^*(s)}(g) \geq Var^M(g)$.

### A.7 CONVERGENCE ANALYSIS OF VRDR

Let $\nabla J(\theta) = \nabla_\theta \mathbb{E}_{p \sim P}[\eta(\pi, p)]$ be the policy gradient. In this section, we present the theorem proposed by Ghadimi & Lan (2013) under the DR setting. It shows that under Assumption 1 that $\nabla J(\theta)$ is Lipschitz continuous w.r.t. policy parameter $\theta$, the variance of gradient estimator dominates the convergence rate.

**Assumption 1.** $\|\nabla J(\theta_1) - \nabla J(\theta_2)\| \leq L\|\theta_1 - \theta_2\|, \quad \forall \theta_1, \theta_2 \in \mathbb{R}^n.$

Applying an arbitrary gradient estimator, the convergence of policy gradient is upper bounded as follows.

**Theorem 3.** *(Ghadimi & Lan, 2013) Under Assumption 1, and assuming that the step size $\alpha_k = \min\{\frac{1}{L}, \frac{1}{\sqrt{Var(g_k)K}}\}$, we have*

$$\mathbb{E}_R \|\nabla J(\theta)\|_2^2 \leq \frac{L(J(\theta^*) - J(\theta_1))}{K/2} + \frac{L \max_k \sqrt{Var(g_k)}}{\sqrt{K}},$$

(40)

where $J(\theta^*)$ is the maximum expected cumulative discounted reward that can be achieved by policy $\pi_{\theta^*}$, $K$ is the maximum number of iterations for policy gradient method, $g_k = g(\theta_k, s, a)$ is the gradient estimator at the $k$-th iteration for $(s, a)$, and

$$\mathbb{E}_R \|\nabla J(\theta)\|_2^2 = \frac{\sum_{k=1}^{K}(\alpha_k - L/2\alpha_k^2)\mathbb{E}\|\nabla J(\theta_k)\|_2^2}{\sum_{k=1}^{K}(\alpha_k - L/2\alpha_k^2)} \tag{41}$$

is the expectation w.r.t. distribution $P_R(\theta_k) = \frac{\alpha_k - L/2\alpha_k^2}{\sum_{k=1}^{K}(\alpha_k - L/2\alpha_k^2)}$ and all the possible randomness.

By holding Eq. (12), we have $\frac{Var^{b^*(s)}(g)}{Var^M(g)} > 1$ and gains a $\kappa = \sqrt{\frac{Var^{b^*(s)}(g)}{Var^M(g)}}$ $(\kappa > 1)$ acceleration on the convergence of policy gradient methods.

## A.8 ANALYSIS OF ADDITIONAL COMPUTATIONAL COST OF VRDR

In DR, the computational cost contains mainly the feed-forward computation and back-propagation computation of the policy network and the value network. For $N$ iterations, $H$ trajectories with length of $L_h$, size of state space $|\mathcal{S}|$ and constant computational cost of feed-forward and back-propagation computation $d_1$, the computational cost of DR is $O(N \cdot H \cdot L_h \cdot |\mathcal{S}| \cdot d_1)$.

The additional computational cost introduced by VRDR is mainly from the clustering phase. Considering the hierachical clustering method in Algorithm 2. The maximum number of outer loop is $H - M < H$. At each outer loop, the computational cost of distance matrix is $m^2 \leq H^2$. Then, let the cost of distance computation in Line 6 of Algorithm 2 be $d_2$, the computational cost is $O(N \cdot H^3 \cdot d_2)$.

Note that compared with the simple computation of distance between two scalars $u_i$ and $u_j$, the computational cost of feed-forward and back-propagation computation are implemented for the entire policy and value networks, both of which contain multiple layers and neurons. Therefore, in practice, $d_1$ is usually multiple order-of-magnitude larger than $d_2$. Thus, under the condition that $L_h$ and $|\mathcal{S}|$ are comparable with $H$, term $O(N \cdot H \cdot L_h \cdot |\mathcal{S}| \cdot d_1)$ will become the dominate term in the overall computational complexity of VRDR, while the additional computational complexity introduced by the clustering phase of VRDR can be neglected.

## A.9 CLUSTERING RESULTS DURING TRAINING

In this subsection, we show the clustering results of three successive clustering updates (iteration $k = nN_c, (n + 1)N_c, (n + 2)N_c$) on Walker2d, Hopper, and Halfcheetah in Figs. 5(a)-5(c).

## A.10 ABLATION STUDY OF VRDR ON CLUSTERING METHODS

In this subsection, we study the influence of different clustering methods on VRDR. Considering that we require to specify the number of clusters in VRDR and the clusters should be generated with similar state values to satisfy Eq. (12), we additionally chose k-means for the comparison to the originally adopted hierachical clustering. The comparison results of different clustering methods are shown in Fig. 6(a) and Fig. 6(b). It can be seen that applying k-means and hierachical clustering methods present similar average scores on Hopper and Halfcheetah.

## A.11 COMPARISON TO COMBINED STATE-DEPENDENT BASELINE

The optimal state/environment baseline can be considered as combined state-dependent baseline, where combined state is composed of original state $s$ and environment parameter $p$, if we treat $p$ as part of state of the MDP. In this subsection, we conduct experiment to evaluate DR with combined state-dependent baseline (DRCS) and the training curves are shown in Fig. 7(a) and Fig. 7(b). The curves of DRCS show that applying regular state-dependent baseline on this combined state degrades the average score during training as compared to DR and VRDR.

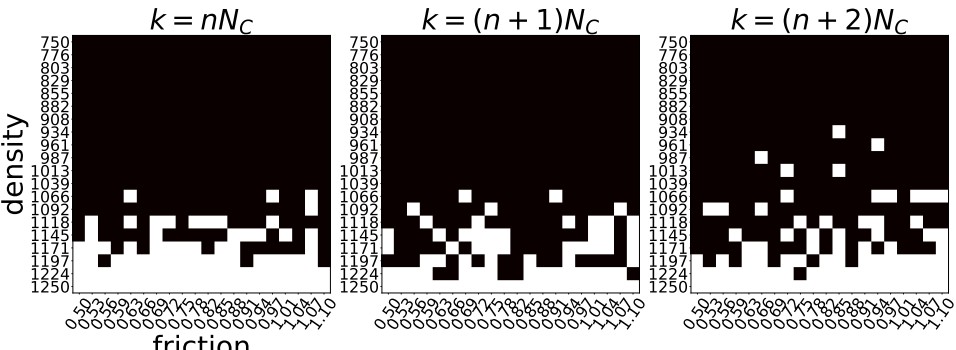

(a) Walker2D

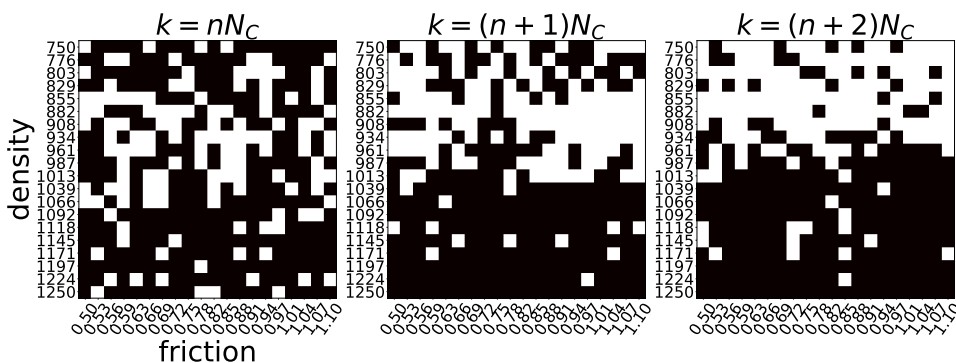

(b) Hopper

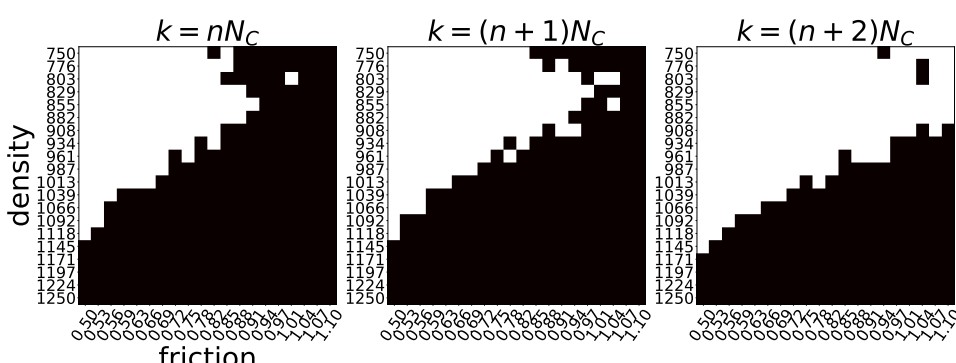

(c) Halfcheetah

Figure 5: Clustering results in three successive clustering update.

## A.12 COMPARISON TO OTHER BASELINE ALGORITHMS

In this subsection, we compare VRDR with other baseline algorithms including DR with meta baseline (DRMB) (Mao et al., 2018; Liu et al., 2019) and Epopt (Rajeswaran et al., 2016). We use PPO with the same setting as in Section 5. In DRMB, a meta baseline is applied to learn the simpler

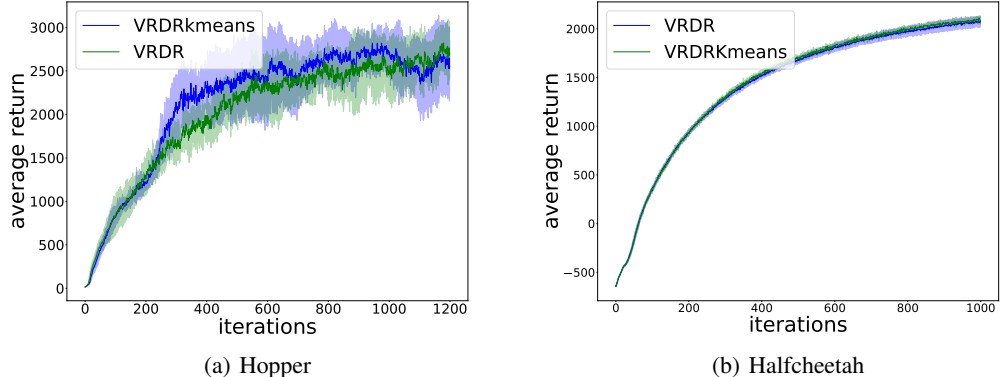

(a) Hopper

(b) Halfcheetah

Figure 6: Training curves of average return achieved by using k-means and hierachical clustering in VRDR.

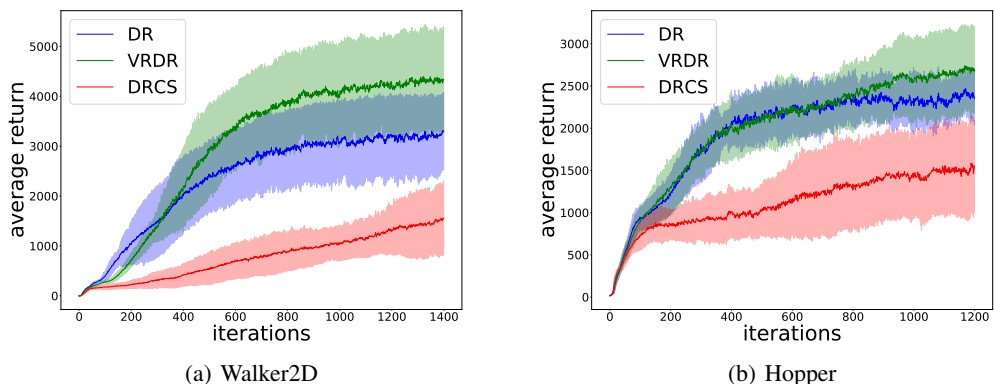

(a) Walker2D

(b) Hopper

Figure 7: Training curves of average return achieved by DR, VRDR and DRCS.

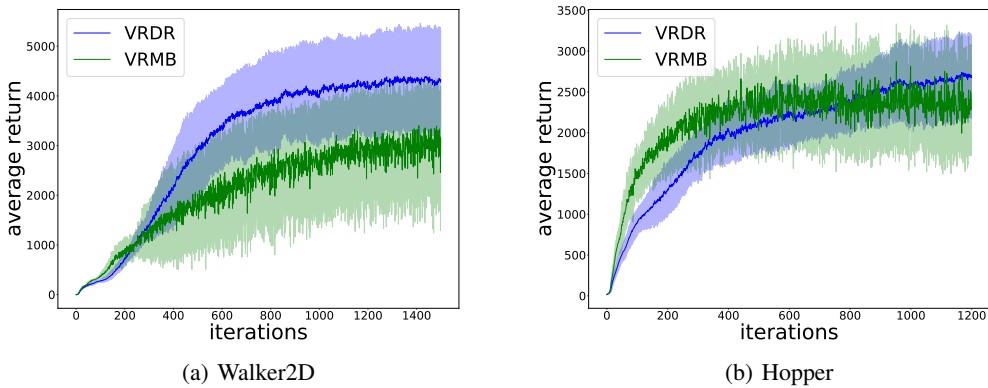

(a) Walker2D

(b) Hopper

Figure 8: Training curves of average return achieved by VRDR and VRMB.

alternative $\mathbb{E}_\pi [Q_\pi(s, a, p)]$ instead of state/environment-dependent baseline $b^*(s, p)$. The training curves are shown in Fig. 8(a) and Fig. 8(b). VRDR shows better average scores than DRMB on Walker2d and a similar average score on Hopper.

In Epopt, we use purely trajectories from the $\alpha$-percentile worst-case environments for training, where we set $\alpha = 5$. The training curves are shown in Fig. 9(a) and Fig.9(b). And it can be seen that VRDR outperforms Epopt on both environments.

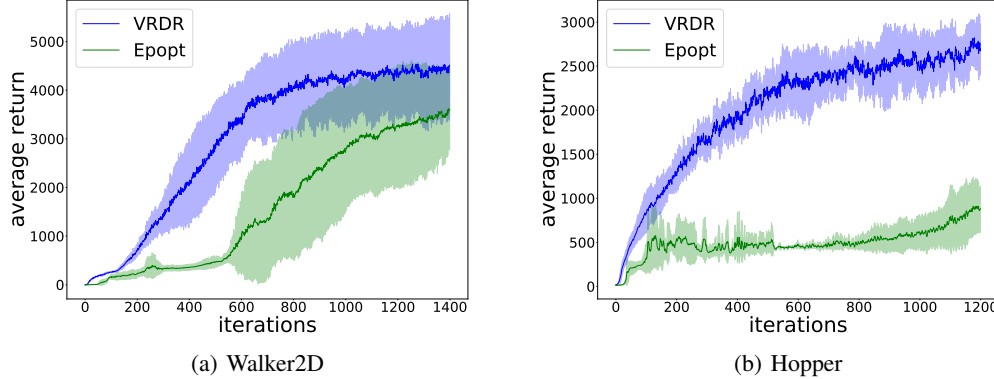

(a) Walker2D                    (b) Hopper

Figure 9: Training curves of average return achieved by VRDR and Epopt.

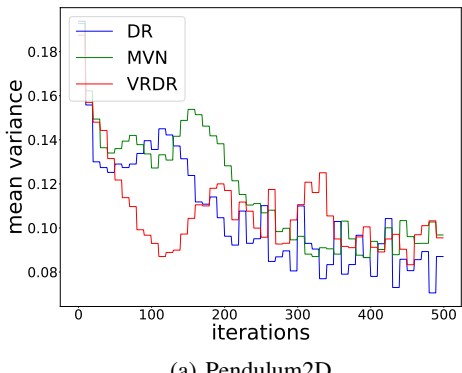

(a) Pendulum2D

Figure 10: Training curves of mean variance of gradient estimates over random seeds achieved by DR, MVN and VRDR.

### A.13 VARIANCE OF GRADIENT ESTIMATES DURING TRAINING

In this subsection, we empirically evaluate the variance of the gradient estimates that are practically achieved by DR, MVN and VRDR on Pendulum2D. It can be seen that at the beginning of training, VRDR can achieve a significantly lower variance of gradient estimates than the other two baselines, which coincides with the faster convergence behavior of VRDR as shown in Fig. 1(e).

### A.14 HIERACHICAL CLUSTERING FOR VRDR

In this subsection, we detail the hierarchical clustering method used in the implementation of VRDR, and summarize it in Algorithm 2.

### A.15 FROM EQ. (12) TO EQ. (11)

In this subsection, we prove that $Var^{\mathcal{P}}(V_\pi(s,p)) \geq Var^{\mathcal{P}_j}(V_\pi(s,p))$ in Eq. (12) is a sufficient condition of $Var^{\mathcal{P}}(Y_p(s)) \geq Var^{\mathcal{P}_j}(Y_p(s))$ in Eq. (11). When the above condition holds, we have

$$Var^{\mathcal{P}}(V_\pi(s,p)) \geq Var^{\mathcal{P}_j}(V_\pi(s,p)), \tag{42}$$

and by multiplying both sides with a non-negative term $\mathbb{E}_a[G(a,s)]$, we have

$$\mathbb{E}_a[G(a,s)]Var^{\mathcal{P}}(V_\pi(s,p)) \geq \mathbb{E}_a[G(a,s)]Var^{\mathcal{P}_j}(V_\pi(s,p)). \tag{43}$$

---

**Algorithm 2** Hierachical Clustering

---

1: **Input** $\Omega = \{\hat{V}_\pi(p_c)\}_{c=1}^H$,
2: **Initialize** number of clusters $M$, current number of clusters $m = H$, prototypes $\{u_i = \hat{V}_\pi(p_i)\}_{i=1}^H$, distance matrix $M_d$
3: **while** $m \geq M$ **do**
4:     **for** $i = 0$ to $m - 1$ **do**
5:         **for** $j = 0$ to $m - 1$ **do**
6:             $M_d(i, j) = |u_i - u_j|$
7:             $M_d(i, j) = M_d(j, i)$
8:         **end for**
9:     **end for**
10:     merge the two clusters $u_{i^*}$ and $u_{j^*}$ having the minimum distance: $u_{i^*} = u_{i^*} \cup u_{j^*}$
11:     $m = m - 1$
12: **end while**
13: **Output** prototypes $\{u_i\}_{i=1}^M$

---

By expanding the LHS with the definition of variance, we have

$$\mathbb{E}_a[G(a,s)]Var^{\mathcal{P}}(V_\pi(s,p)) \tag{44}$$

$$=\mathbb{E}_a[G(a,s)]\sum_{i=1}^{|\mathcal{P}|} P(p_i|s)\Big[\mathbb{E}_{\mathcal{P}}[V_\pi(s,p)] - V_\pi(s,p_i)\Big]^2$$

$$=\frac{1}{\mathbb{E}_a[G(a,s)]}\sum_{i=1}^{|\mathcal{P}|} P(p_i|s)\Big[\mathbb{E}_a[G(a,s)]\mathbb{E}_{\mathcal{P}}\Big[\mathbb{E}_a\left[Q_\pi(s,a,p)\right]\Big] - \mathbb{E}_a[G(a,s)]\mathbb{E}_a\left[Q_\pi(s,a,p_i)\right]\Big]^2$$

$$=\frac{1}{\mathbb{E}_a[G(a,s)]}\sum_{i=1}^{|\mathcal{P}|} P(p_i|s)\Big[\mathbb{E}_{\mathcal{P}}\Big[\mathbb{E}_a\left[G(a,s)Q_\pi(s,a,p)\right]\Big] - \mathbb{E}_a\left[G(a,s)Q_\pi(s,a,p_i)\right]\Big]^2$$

$$=\frac{Var^{\mathcal{P}}(Y_p(s))}{\mathbb{E}_a[G(a,s)]},$$

where the third equality can be obtained by using the condition $\mathbb{E}_a[G(a,s)Q_\pi(s,a,p)] = \mathbb{E}_a[G(a,s)]\mathbb{E}_a[Q_\pi(s,a,p)]$. Similarly, we can obtain $\mathbb{E}_a[G(a,s)]Var^{\mathcal{P}_j}(V_\pi(s,p)) = \frac{Var^{\mathcal{P}_j}(Y_p(s))}{\mathbb{E}_a[G(a,s)]}$. Hence, from the inequality in Eq. (43), we have $Var^{\mathcal{P}}(Y_p(s)) \geq Var^{\mathcal{P}_j}(Y_p(s))$, and thus $Var^{b^*(s)}(g) \geq Var^M(g)$ according to Eq. (10). Namely, the improvement of variance reduction by applying subspaces generated by following Eq. (12) can also be quantified by Eq. (10).

