# OpenReview forum: "Variance Reduced Domain Randomization for Policy Gradient"
_ICLR.cc/2022/Conference — ICLR 2022 Submitted_

### Official Review · Reviewer_uerm · 2021-11-02

**Correctness:** 3
**Technical Novelty And Significance:** 2
**Empirical Novelty And Significance:** 2
**Recommendation:** 5
**Confidence:** 4

**Main Review:**

### Strengths ###
- It’s great to see that the proposed algorithm demonstrates good generalizability compared to other methods in Figure 2. I have a question about a plot though. In Figure 2.(a), the most left subplot, the DR’s score is no higher than 1400. However, in Figure 1 (a), the mean of the DR curve is around 3000. Did I interpret the plots correctly?
- Domain randomization is commonly used in practice to train good policies using simulators. The problem that the submission tries to address is relevant to the community.
- I appreciate the efforts of deriving and analyzing the practical algorithms based on theoretical development, although it would be better to analyze the variance reduction of the practical baseline proposed.

### Concerns ###
- The novelty of the submission is limited. As authors noted, the baseline in Section 3.2 is a special case of the input-driven baseline derived by Mao et al. (2018). And it's not discussed how proposed method is different from Liu et al.’ s per-task control variate.
   - If I understand it correctly, the environment parameters can be treated as part of the state of the MDC. The combined state (original state + environment parameter)-dependent baselines can be learned just as regular state-dependent baselines based on function approximation without additional techniques, such as the subspace clustering.
  - In Algorithm 1, clusters are formed based on Q values, but the nearest neighbors are found based on environment parameter. This inconsistency is a bit odd to me.

- There may be a bug in the theoretical analysis. I believe the expectation of environment parameters need to be conditioned on s, just as the action distribution is conditioned on s (policy \pi). This includes E_P in Eq. (4), corollary 1, and Theorem 2.
- The clarity of the paper can be improved.
  - The math notations are sometimes confusing and math arguments are not always precise. Examples:
      - On the fifth line after Eq. (1), the meaning of equation E_{P, \mu, \pi}[g] = .. := E[g], and the meaning of g in Eq. (9) are not very clear.
      - The symbol “|” in probability represents conditioning. However, it seems to represent parameterized functions, for example,  \eta(\pi |p), b(s|p_i), b^*(s|P_j)
     - In Theorem 2, "i.f.f.", the condition (Eq. (11)) seems to be sufficient but not necessary.
     - “can obtain the minimum variance for DR” right above Section 4. I think Corollary 1 shows that state/environment-dependent baseline reduces more variance than state-dependent baseline, but I don’t think this implies state/environment-dependent baseline can obtain the minimum variance.
  - The algorithm requires more explanation, such as, how the baselines are updated in Line 14, how the hierarchical cluster method works. And more description about relabel may help.

### Other comments ###
- Multiple symbols are randomly written in Italic or non-italic font, p in the second before the last line on Page 2 and a in the fifth before the last line in Section 2.
-The meaning of N_C and N in Line 3, Algorithm 1.
- What does $\kappa = (\kappa > 1)$ represent, right above Section 5?
-  {p_c}_{c=1}^M  on the fourth line after Algorithm 1 box.  Should it be {p_c}_{c=0}^{H-1} instead?
- hve -> have in the middle of page 9.

### After rebuttal ###
I really appreciate that the authors incorporated the comments in such a short amount of time. Although I think the clustering idea and experiments are interesting, due to the amount of changes made, I am slightly leaning toward suggesting the authors take some more time improving the manuscript and submit it to future conferences. For example, I think the experiment section can be improved with the new experiments (and discussions on the results) and more analysis on the proposed algorithm (e.g. how b* in the second paragraph on Page 5 is related to the algorithm proposed (we can not sample from (P(p|s)) and quantify the correlation between G(a, s) and Q(s, a, p)).


**Summary Of The Paper:**

This paper tackles the variance of policy gradient due to the domain randomization used in RL in simulations. The authors prove that the policy gradient variance can be further reduced by learning a state-dependent baseline for each environment parameter compared to state-dependent baselines. The authors then develop a practical algorithm based on the analysis and analyze the properties of the algorithm. The algorithm is implemented and tested on six robot control tasks. It consistently accelerates policy training.

**Summary Of The Review:**

Because my concerns outweigh the strengths now, I am leaning towards rejecting the paper.

---

> ### Author Response · Authors · 2021-11-15
> **Initial Response to Reviewer uerm Part 1**
>
> We would like to thank the reviewer for providing the detailed comments. Here, we just wanted to provide our initial response to these comments, and possibly discuss with the reviewer about our revision plan. The final formal response and revised manuscript will be submitted at the end of the discussion period.
>
> **Comment 1**: “I have a question about a plot though. In Figure 2.(a), the most left subplot, the DR’s score is no higher than 1400. However, in Figure 1 (a), the mean of the DR curve is around 3000. Did I interpret the plots correctly?”
>
> **Response**: Thanks for the careful reading. It was a mistake of our plotting and we will correct it in the revised version.
>
> **Comment 2**: “I appreciate the efforts of deriving and analyzing the practical algorithms based on theoretical development, although it would be better to analyze the variance reduction of the practical baseline proposed.”
>
> **Response**: We have analyzed the variance difference of the proposed practical baseline compared to the optimal state-dependent baseline in Theorem 2. Since VRDR makes the inequality specified by Eq. (11) hold, the variance reduction of the proposed practical baseline is guaranteed to be greater than and equal to the optimal state-dependent baseline, where the quantity of reduced variance is specified by Eq. (10).
>
> **Comment 3**: “how proposed method is different from Liu et al.’ s per-task control variate”.
>
> **Response**: Liu et al.‘s per-task control variate is essentially equivalent to the multi-value-network (MVN) proposed by Mao et al., where both methods proposed to maintain a value network as baseline for each task (Liu et al.) or input sequence (Mao et al.). Per-task control variate is constructed by action-dependent baseline as proposed by Wu et al., while the state-dependent baseline is considered in MVN. In this paper, we consider state-dependent in DR, MVN and VRDR for a fair comparison and will consider incorporating action-dependent baseline as our future work. Besides, the meta control variates-based method proposed by Liu et al. is technically the same as the meta input-dependent baseline by Mao et al.. We are now conducting experiments to evaluate the performance of the meta baseline in our robot simulation environments and the results will be provided later.
>
> **References**:
>
> Hongzi Mao, Shaileshh Bojja Venkatakrishnan, Malte Schwarzkopf, and Mohammad Alizadeh.
> Variance reduction for reinforcement learning in input-driven environments. In International
> Conference on Learning Representations, 2018.
>
> Cathy Wu, Aravind Rajeswaran, Yan Duan, Vikash Kumar, Alexandre M Bayen, Sham Kakade,
> Igor Mordatch, and Pieter Abbeel. Variance reduction for policy gradient with action-dependent factorized baselines. arXiv preprint arXiv:1803.07246, 2018
>
> Hao Liu, Richard Socher, and Caiming Xiong. Taming maml: Efficient unbiased metareinforcement learning. In International Conference on Machine Learning, pp. 4061–4071. PMLR, 2019.
>
> **Comment 4**: “If I understand it correctly, the environment parameters can be treated as part of the state of the MDC. The combined state (original state + environment parameter)-dependent baselines can be learned just as regular state-dependent baselines based on function approximation without additional techniques, such as the subspace clustering.”
>
> **Response**: We used to conduct related experiments, where environment parameters are treated as extra state of the MDP and the corresponding value function takes combined state as input. The results showed that this simple treatment would degrade the average score during training. This part of results will be provided later in the revised version.
>
> **Comment 5**: “In Algorithm 1, clusters are formed based on Q values, but the nearest neighbors are found based on environment parameter. This inconsistency is a bit odd to me.”
>
> **Response**: The clusters are formed based on estimated Q value as shown in Lines 5 and 6. However, the labels of two similar clusters in different clustering computation would be different along the training iterations. To avoid this, we relabel the new prototype $\bf{u}'$ generated by clustering method to the labels $\bf l=$ {$ l_i $} that are nearest to the old prototypes as shown in Line 8 in Algorithm 1.
>
> **Comment 6**: “There may be a bug in the theoretical analysis. I believe the expectation of environment parameters need to be conditioned on s, just as the action distribution is conditioned on s (policy \pi). This includes E_P in Eq. (4), corollary 1, and Theorem 2.”
>
> **Response**: The expectation of environment parameters conditioned on $s$ means that $p\sim P(p\vert s)$, whereas knowing $s$ won't change the sampling distribution $P(p)$ in the DR setting. Therefore, the expectation of environment parameters is not conditioned on $s$.

---

> > ### Author Response · Authors · 2021-11-15
> > **Initial Response to Reviewer uerm Part 2**
> >
> > **Comment 7**:  “On the fifth line after Eq. (1), the meaning of equation E_{P, \mu, \pi}[g] = .. := E[g], and the meaning of g in Eq. (9) are not very clear.”
> >
> > **Response**: We would like to thank the reviewer for the careful reading, and will denote the gradient estimators that subtracts constant, state-dependent and state/environment baseline by $g_c$, $g_s$ and $g_p$ in the revised version for clarity.
> >
> > **Comment 8**:   “The symbol “|” in probability represents conditioning. However, it seems to represent parameterized functions, for example, \eta(\pi |p), b(s|p_i), b^*(s|P_j)”
> >
> > **Response**: Thanks for pointing out this ambiguity. We will make it clearer in the revised version.
> >
> > **Comment 9**:  "In Theorem 2, "i.f.f.", the condition (Eq. (11)) seems to be sufficient but not necessary."
> >
> > **Response**: Thanks for the careful reading. It should have been a sufficient condition and we will correct it in the revised version.
> >
> > **Comment 10**:  "can obtain the minimum variance for DR” right above Section 4. I think Corollary 1 shows that state/environment-dependent baseline reduces more variance than state-dependent baseline, but I don’t think this implies state/environment-dependent baseline can obtain the minimum variance."
> >
> > **Response**:  In Appendix A.1, we have shown that the variance optimization problem in Eq. (20) was convex w.r.t. state/environment-dependent baseline $b(s, \mathcal{P})$. Besides, state-dependent baseline can be considered as a special case of $b(s, \mathcal{P})$, where $b(s, \mathcal{P})$ is invariant across different environment. Constant baseline can be considered as special case of state-dependent baseline $b(s)$ that is invariance across different state. Hence, the optimal state/environment-dependent baseline $b^*(s,\mathcal{P})$ can obtained by solving the convex problem in Eq. (20), with the minimal variance for DR over all the baselines that are state/environment-dependent, state-dependent and constant.
> >
> > **Comment 11**: "The algorithm requires more explanation, such as, how the baselines are updated in Line 14, how the hierarchical cluster method works. And more description about relabel may help."
> >
> > **Response**: Thanks for the suggestion. We will provide more detailed description for Algorithm 1 in the revised version.
> >
> > **Comment 12**: "Multiple symbols are randomly written in Italic or non-italic font, p in the second before the last line on Page 2 and a in the fifth before the last line in Section 2."
> >
> > **Response**: Thanks for the careful reading. We use non-italic symbol to represent random variable as recommended by formatting instructions for ICLR 2022. Hence p and a are both random variables here.
> >
> > **Comment 13**: "The meaning of N_C and N in Line 3, Algorithm 1."
> >
> > **Response**:  We will define $N_C$ and $N$ in the revised version at the beginning of Algorithm 1.
> >
> > **Comment 14**: "What does \kappa=(\kappa>1) represent, right above Section 5?"
> >
> > **Response**: $\kappa$ is the square root of gradient estimator's variance ratio between applying $b^*(s)$ and $b^M(s,\mathcal{P})$ as defined in Appendix A.6. By applying the theorem of convergence of stochastic gradient descent method provided by Ghadimi, we obtained Theorem 3 and showed that VRDR could gain a $\kappa$ acceleration on the convergence compared to the direct application of optimal state-dependent baseline for DR.
> >
> > **Reference**:
> >
> > Saeed Ghadimi and Guanghui Lan. Stochastic first-and zeroth-order methods for nonconvex stochastic programming. SIAM Journal on Optimization, 23(4):2341–2368, 2013.
> >
> > **Comment 15**: "{p_c}{c=1}^M on the fourth line after Algorithm 1 box. Should it be {p_c}{c=0}^{H-1} instead?"
> >
> > **Response**: Thanks for the careful reading and we will correct it in the revised version.
> >
> > **Comment 16**: “hve -> have in the middle of page 9.”
> >
> > **Response**: Thanks for the careful reading and we will correct it in the revised version.

---

> > > ### Comment · Reviewer_uerm · 2021-11-20
> > > **Follow up questions**
> > >
> > > Thanks to the authors for the response.
> > >
> > > Comment 2 I am sorry that I was not clear. I wanted to say that VRDR does not estimate Var(Y) as proposed in Theorem 2 but estimates Var(Q) instead. Although there’s some justification around Eq. (12), more formal justification would help.
> > >
> > > Comment 6 I still think that’s a bug. There’s a joint distribution of s and p. The generating process is (1)  sampling p uniformly, and (2) executing some policy and sampling s. E_P[Y_P(s)] in Eq. (1) is a function of s, and E_P is an expectation over the *conditional* distribution P given s. The conditional distribution is defined by the joint distribution of s and p. For example, if an environment p1 does not visit state s1 at all, the prob(p1|s1) = 0.
> > >
> > > Comment 10 I agree that b*(s, p) is optimal among state-environment-dependent baselines, but there are other baselines that are not state-environment-dependent baselines, e.g. state-action dependent baselines. Therefore, I think it’s improper to claim that “..b*(s, p) can obtain the minimum variance for DR.”.

---

> > > > ### Author Response · Authors · 2021-11-22
> > > > **Response to Follow up questions Part 1**
> > > >
> > > > **Comment 17**: "Comment 2 I am sorry that I was not clear. I wanted to say that VRDR does not estimate Var(Y) as proposed in Theorem 2 but estimates Var(Q) instead. Although there’s some justification around Eq. (12), more formal justification would help."
> > > >
> > > > **Response**: First of all, we would like to correct the condition $E_{s,a}[G(a, s)Q_{\pi}(s,a \vert p)] = E_{s,a}[G(a, s)]E_{s,a}[Q_{\pi}(s,a \vert p)]$, which we originally stated in the first line above Eq. (12), to $E_{a}[G(a, s)Q_{\pi}(s,a , p)] = E_{a}[G(a, s)]E_{a}[Q_{\pi}(s,a , p)]$. Please also note that we have corrected the usage of symbol "$\vert$'' as have been pointed out in Comment 8, and changed $Q_{\pi}(s,a \vert p)$ to $Q_{\pi}(s,a,p)$ here and in the revised version.
> > > >
> > > > 1) In the revised version, we estimate $Var(V_{\pi}(s, p))$ instead of $Var(Y_p(s))$ based on the above condition $E_{a}[G(a, s)Q_{\pi}(s,a , p)] = E_{a}[G(a, s)]E_{a}[Q_{\pi}(s,a , p)]$. This condition holds when the inner product of log policy $G(s,a)$ is loosely correlated to the $Q$-value function. We showed in the footnote on Page 5 an illustrating example where this condition holds. And the same condition is also applied by Mao et al. (2018) and Wu et al. (2018) to derive their practical input-driven baseline and action-dependent baseline, respectively, according to their theoretical optimums.
> > > >
> > > > 2) We then prove that $Var^{P}(V_{\pi}(s, p)) \geq Var^{P_j}(V_{\pi}(s, p))$ is a sufficient condition of $Var^{P}(Y_p(s)) \geq Var^{P_j}(Y_p(s))$ in Eq. (11). When the above condition holds, we have
> > > > \begin{align}
> > > > 	Var^{P}(V_{\pi}(s, p)) \geq Var^{P_j}(V_{\pi}(s, p)),
> > > > \end{align}
> > > > and by multiplying both sides with a non-negative term $E_{a}[G(a, s)]$, we have
> > > > \begin{align}
> > > >  E_{a}[G(a, s)]
> > > > Var^{P}(V_{\pi}(s, p)) \geq E_{a}[G(a, s)]Var^{P_j}(V_{\pi}(s, p)).
> > > > \end{align}
> > > > By expanding the LHS with the definition of variance, we have
> > > > \begin{align}
> > > > E_{a}[G(a, s)]
> > > > Var^{P}(V_{\pi}(s, p))=E_{a}[G(a, s)] \sum_{i=1}^{\vert P \vert } P(p_i \vert s) \Big [ E_{P}[V_{\pi}(s,p)] - V_{\pi}(s,p_i) \Big ]^2
> > > > = \frac{1}{E_{a}[G(a, s)]} \sum_{i=1}^{\vert P \vert } P(p_i \vert s) \biggl [ E_{a}[G(a, s)] E_{P} \Big [ E_{a}\left[Q_{\pi}(s,a, p) \right] \Big ] -  E_{a}[G(a, s)] E_{a}\left[ Q_{\pi}(s,a, p_i) \right] \biggr ]^2
> > > > = \frac{1}{E_{a}[G(a, s)]}\sum_{i=1}^{\vert P \vert } P(p_i \vert s) \biggl [ E_{P} \Big [E_{a}\left[G(a, s) Q_{\pi}(s,a, p) \right] \Big ]  - E_{a}\left[G(a, s) Q_{\pi}(s,a, p_i) \right] \biggr ] ^2
> > > > = \frac{Var^{P}(Y_p(s))}{E_{a}[G(a, s)]},
> > > > \end{align}
> > > > where the third equality can be obtained by using the above condition. Similarly, we can obtain $E_{a}[G(a, s)]Var^{P_j}(V_{\pi}(s, p))=\frac{Var^{P_j}(Y_p(s))}{E_{a}[G(a, s)]}$. Hence, from the above inequality, we have $Var^{P}(Y_p(s)) \geq Var^{P_j}(Y_p(s))$, and thus $Var^{b^*(s)}(g) \ge Var^{M}(g)$ according to Eq. (10). Namely, the improvement of variance reduction by applying subspaces generated by following Eq. (12) can also be quantified by Eq. (10).
> > > >
> > > > **References**:
> > > > Hongzi Mao, Shaileshh Bojja Venkatakrishnan, Malte Schwarzkopf, and Mohammad Alizadeh. Variance reduction for reinforcement learning in input-driven environments. In International Conference on Learning Representations, 2018.
> > > >
> > > > Wu, C., Rajeswaran, A., Duan, Y., Kumar, V., Bayen, A. M.,Kakade, S., Mordatch, I., and Abbeel, P. Variance reduction
> > > > for policy gradient with action-dependent factorized baselines. In International Conference on Learning Representations, 2018.

---

> > > > ### Author Response · Authors · 2021-11-22
> > > > **Response to Follow up questions Part 2**
> > > >
> > > > **Comment 18**: "Comment 6 I still think that’s a bug. There’s a joint distribution of s and p. The generating process is (1) sampling p uniformly, and (2) executing some policy and sampling s. E_P[Y_P(s)] in Eq. (1) is a function of s, and E_P is an expectation over the conditional distribution P given s. The conditional distribution is defined by the joint distribution of s and p. For example, if an environment p1 does not visit state s1 at all, the prob(p1|s1) = 0."
> > > >
> > > > **Response**: Thanks for your carefully reading. There is indeed a bug in Eq. (4), Corollary 1, and Theorem 2. Since Corollary 1 and Theorem 2 were proved by applying the relationship in Eq. (4), we first provide here the correct derivation of Eq. (4) by solving the variance minimization problem as formulated in Eq. (16):
> > > > \begin{align}
> > > >      \min_{b(s)} F(b(s)) = E_{P, \mu^p_{\pi},\pi} \Big [ G(a, s) [Q_{\pi}(s,a, p)-b(s)]^2 \Big ].
> > > > \end{align}
> > > >
> > > > Following the generating process of joint distribution of $p$, $s$ and $a$, $P(p,s,a)=P(p)P(s\vert p)P(a \vert s, p) = P(p)\mu_{\pi}(s\vert p)\pi(a \vert s)$, where the last equality is because that policy $\pi$ is constructed without knowing $p$. The RHS of the above equation can be expanded as:
> > > > \begin{align}
> > > >     E_{p\sim P,\mu^p_{\pi}} \left[ E_{\pi}\Big [ G(a, s) [Q_{\pi}(s,a, p)-b(s, p)]^2 \Big ] \right ]
> > > >     = \sum_{i=1}^{\vert P \vert} P(p_i) \sum_{s_j}\mu_{\pi}(s_j\vert p_i) E_{\pi} \Big [ G(a, s_j) [Q_{\pi}(s_j,a, p_i)-b(s_j, p_i)]^2 \Big ]
> > > >     =\sum_{j=1}^{\vert \mathcal{S} \vert} \sum_{i=1}^{\vert P \vert} P(p_i) \mu_{\pi}(s_j\vert p_i) E_{\pi} \Big [ G(a, s_j) [Q_{\pi}(s_j,a, p_i)-b(s_j, p_i)]^2 \Big ].
> > > > \end{align}
> > > >
> > > > By letting the first-order partial derivative equal to zero:
> > > > \begin{align}
> > > >     \frac{\partial F(b(s))}{\partial b(s)}=-2\sum_{j=1}^{\vert \mathcal{S} \vert} \sum_{i=1}^{\vert P \vert} P(p_i) \mu_{\pi}(s_j\vert p_i)E_{\pi}\Big [ G(a, s_j) [Q_{\pi}(s_j,a, p_i)-b(s_j)] \Big ]=0,
> > > > \end{align}
> > > > where $b(s)=\{ b(s_j) \}$, we have:
> > > > \begin{align}
> > > >     \sum_{i=1}^{\vert P \vert} P(p_i) \mu_{\pi}(s_j\vert p_i)E_{\pi}\Big [ G(a, s_j) [Q_{\pi}(s_j,a, p_i)-b(s_j)] \Big ]=0,\quad \text{for each $b(s_j)$.}
> > > > \end{align}
> > > > It can be further written as
> > > > \begin{align}
> > > >     \sum_{i=1}^{\vert P \vert} P(p_i, s_j)E_{\pi}\Big [ G(a, s_j) [Q_{\pi}(s_j,a, p_i)] \Big ]&=\sum_{i=1}^{\vert P \vert} P(p_i, s_j)E_{\pi}\Big [ G(a, s_j) [b(s_j)] \Big ],
> > > > \end{align}
> > > > and
> > > > \begin{align}
> > > > \sum_{i=1}^{\vert P \vert} P(s_j) P(p_i\vert s_j)E_{\pi}\Big [ G(a, s_j) [Q_{\pi}(s_j,a, p_i)] \Big ]=\sum_{i=1}^{\vert P \vert} P(s_j)  P(p_i\vert s_j)E_{\pi}\Big [ G(a, s_j) \Big ]b(s_j),
> > > > \end{align}
> > > > where $P(s_j) $ can be further canceled on both sides. We can then get the optimal state-dependent baseline for $s_j$ as
> > > > \begin{align}
> > > >     b^*(s_j) = \frac{\sum_{i=1}^{\vert P \vert} P(p_i\vert s_j) E_{\pi}\Big [ G(a, s_j) [Q_{\pi}(s_j,a, p_i)] \Big ]}{\sum_{i=1}^{\vert P \vert} P(p_i\vert s_j)E_{\pi}\Big [ G(a, s_j) \Big ]} = \frac{E_{P(p\vert s_j)}E_{\pi}\Big [ G(a, s_j) [Q_{\pi}(s_j,a, p)] \Big ]}{E_{\pi}\Big [ G(a, s_j) \Big ]}.
> > > > \end{align}
> > > >
> > > > For a continuous state space, we have
> > > > \begin{align}
> > > >     b^*(s) = \frac{E_{P(p\vert s)}E_{\pi}\Big [ G(a, s) [Q_{\pi}(s,a, p)] \Big ]}{E_{\pi}\Big [ G(a, s) \Big ]},
> > > > \end{align}
> > > > which should be the correct Eq. (4) after revision.
> > > >
> > > > Here  $E$ is used to represent $\mathbb{E}$ and $\vert P \vert$ is used to represent $\vert \mathcal{P} \vert$.
> > > >
> > > > **Comment 19**: "Comment 10 I agree that b*(s, p) is optimal among state-environment-dependent baselines, but there are other baselines that are not state-environment-dependent baselines, e.g. state-action dependent baselines. Therefore, I think it’s improper to claim that “..b*(s, p) can obtain the minimum variance for DR.”."
> > > >
> > > > **Response**: Thanks for your suggestion. We will reclaim it as "...$b^*(s,p)$ can obtain the minimum variance for DR within the baselines that consider both state and environment parameters'".

---

> ### Author Response · Authors · 2021-11-23
> **Final Response and Summary of Changes to Reviewer uerm**
>
> We would like to thank again the reviewer for providing the detailed comments and discussing with us on some important concerns. To address these comments, we have also made corresponding changes, as highlighted in blue, in the rebuttal version of our paper. Here, we would like to summarize the major revisions that we have made in the rebuttal version.
>
> **1.	Plotting mistake in Figure 2(a)**: we have replaced it with corrected figure in the same place.
>
> **2.	Difference from Liu et al.’s per-task control variate**: We have conducted the experiments to evaluate the performance of meta baseline and the results are shown as VRMB in Fig. 8(a) and Fig. 8(b) in Appendix A.12.
>
> **3.	Derivation of Theorem 2**: We have re-derived Theorem 2 in Appendix A.5, including providing detailed derivation of Eq. (10), and that Eq. (11) is a sufficient condition of Eq. (10). In addition, in Response to Comment 17 and in Appendix A.15, we have shown that Eq. (12) is a sufficient condition of Eq. (11).
>
> **4.	Comparison to combined state-dependent baseline**: We have conducted the experiments to evaluate DR with combined state-dependent baseline (DRCS), presenting the training curves of average score in Fig. 7(a) and Fig. 7(b) in Appendix A.11. It can be seen that DRCS generally performs worse than DR and VRDR.
>
> **5.	Evaluation of variance of gradient estimates during training**: We have provided the practical training curves, presenting the variance of gradient estimates of DR, MVN and VRDR during training on Pendulum2D in Fig. 10. It can be seen from Fig. 10 that the beginning of training, VRDR can achieve a significantly lower variance of gradient estimates than the other two baselines, which coincides with the faster convergence behavior of VRDR as shown in Fig. 1(e).
>
> **6.	Bug in the theoretical analysis**:
>
> 1) We have re-derived the optimal state-dependent baseline in Appendix A.1, where Eq. (4) has been corrected from taking the expectation over P(p) to over P(p|s).
>
> 2) We have corrected E_P[Y_P(S)] to E_{P(p|s)}[Y_P(S)] in corollary 1.
>
> 3) We have provided a more clear derivation of Theorem 2 in Appendix A.5 based on the corrected Eq. (4).
>
> **7.  Clarity of Algorithm 1**: We have modified Algorithm 1 for more clarity and added Algorithm 2 to describe the hierarchical clustering methods that we have used in practice to implement Algorithm 1. Besides, we have added explanation on how the baselines are updated and the relabeling in Section 4 under Algorithm 1.
>
> **8.	Other concerns**: Please refer to our initial responses to other comments.

---

### Official Review · Reviewer_kFGn · 2021-11-02

**Correctness:** 4
**Technical Novelty And Significance:** 2
**Empirical Novelty And Significance:** 2
**Recommendation:** 5
**Confidence:** 5

**Main Review:**

This manuscript is a re-derivation of the control variate given that the randomness in the environment is partially artificial (domain randomization). The control variate induces a baseline that depends not only on the state but also on the environment. This term unsurprisingly has some traceability issue in practice and then the authors provide a divide-and-conquer idea to partially address it. Experiments show that this method is marginally better than vanilla domain randomization.

Pros:
1. The paper tackles an interesting problem: how to reduce the variance and improve sample efficiency during the training of DR by developing a better baseline.
2. The paper is clearly written in general.

Discussion:
1. Theorem 1 is not useful. The term can be both positive and negative. I don't think control variates in RL has guarantees in variance reduction so I would suggest removing the theorem.
2. The proposed method is based on the premise that the environment parameter is known and used to calculate the clustering prototypes during the training, which may not be valid for the real setting where the agent only can observe the environment.
3. The convergence improvement in Figure 1 is not significant for some tasks like Pendulum and Pendulum2D yet with the additional cost of the clustering process.
4. The paper uses the hierarchical cluster method to partition the environment space. Does the cluster method influence the experimental results? Why are other cluster methods like k-means not considered? And choosing a proper clustering interval seems nontrivial with grid search since the values in Figure 4 fall into different ranges for different tasks.
5. The parameter Nc is used without definition in Alg 1.
6. In section 5.2, the authors learn 15 policies for each algorithm on testing environments to test the generalization to the unseen environment. I guess that the authors wanted to express that they use the trained policies on testing environments directly.
7. The paper uses the uniform domain randomization as a baseline, which is not the proposed method in [Mehta et al., 2020] (cited by the paper). Why does the paper not compare to the active domain randomization algorithm developed in [Mehta et al., 2020]?


**Summary Of The Paper:**

This paper tackles the high variance problem caused by the randomization of environments for estimating the policy gradients. The idea is to derive a bias-free and state/environment-dependent optimal baseline for domain randomization. The authors further develop a VRDR method by dividing the entire environment space into subspaces and estimating the state/subspace-dependent baseline.

**Summary Of The Review:**

This is a marginal improvement of domain randomization with marginal improvement in experiments.

---

> ### Author Response · Authors · 2021-11-15
> **Initial Response to Reviewer kFGn Part 1**
>
> We would like to thank the reviewer for providing the detailed comments. Here, we just wanted to provide our initial response to these comments, and possibly discuss with the reviewer about our revision plan. The final formal response and revised manuscript will be submitted at the end of the discussion period.
>
> **Comment 1**: “Theorem 1 is not useful. The term can be both positive and negative. I don't think control variates in RL has guarantees in variance reduction so I would suggest removing the theorem.”
>
> **Response**: Theorem 1 quantifies the variance reduction improvement for gradient estimator by incorporating the optimal state/environment-dependent baseline $b^*(s,\mathcal{P})$  compared to an arbitrary state-dependent baseline $b(s)$. The quantity of improvement is the RHS term of Eq. (7), which is the expectation of the square of random variable and would always be greater than and equal to zero. Hence the variance reduction for policy gradient is guaranteed theoretically.
>
> **Comment 2**: “The proposed method is based on the premise that the environment parameter is known and used to calculate the clustering prototypes during the training, which may not be valid for the real setting where the agent only can observe the environment.”
>
> **Response**: We focus on tasks that domain randomization (DR) aims to tackle (e.g., robot control), where simulator is accessible and can generate random environment parameters in source domain. We also demonstrate the generalization ability on unseen testing environments in Section 5.2, where the settings of environment parameter are not observable for the RL agent.
>
> **Comment 3**: “The convergence improvement in Figure 1 is not significant for some tasks like Pendulum and Pendulum2D yet with the additional cost of the clustering process.”
>
> **Response**: Thanks for pointing out this issue. We are currently conducting experiments to evaluate the variance of the gradient estimator of DR, MVN and VRDR, and trying to build up an empirically justified relation between the variance of gradient estimator and algorithm’s convergence performance. The results will be provided later when the experiments are done.
>
> **Comment 4**: “The paper uses the hierarchical cluster method to partition the environment space. Does the cluster method influence the experimental results? Why are other cluster methods like k-means not considered? And choosing a proper clustering interval seems nontrivial with grid search since the values in Figure 4 fall into different ranges for different tasks.”
>
> **Response**: Thanks for this suggestion. Since hierarchical clustering method groups environment parameters with similar $Q-$ value, it satisfies Eq. (12). As have been suggested by the reviewer, it is also important to validate different clustering methods’ impact on the VRDR’s performance as an ablation study. We are currently conducting experiments to study the influence of different cluster methods to VRDR. Considering that we require to specify the number of clusters in VRDR, we additionally chose k-means for the comparison to our current choice of hierarchical clustering. The results will be provided later when the experiments are done.
>
> **Comment 5**: “The parameter Nc is used without definition in Alg 1.”
>
> **Response**: Thanks for pointing out that, and we will define it at the beginning of Algorithm 1 in the revised version.

---

> > ### Author Response · Authors · 2021-11-15
> > **Initial Response to Reviewer kFGn Part 2**
> >
> > **Comment 6**:  “In section 5.2, the authors learn 15 policies for each algorithm on testing environments to test the generalization to the unseen environment. I guess that the authors wanted to express that they use the trained policies on testing environments directly.”
> >
> > **Response**: We learned 15 policies for each algorithm on training environments with different random seeds and applied the trained policies on corresponding testing unseen environments. The generalization performance for each algorithm is measured by the average score over 15 policies. Thanks for pointing out this ambiguity, and we will clarify in the revised version.
> >
> > **Comment 7**: “The paper uses the uniform domain randomization as a baseline, which is not the proposed method in [Mehta et al., 2020] (cited by the paper). Why does the paper not compare to the active domain randomization algorithm developed in [Mehta et al., 2020]?”
> >
> > **Response**: Uniform domain randomization has been broadly used in simulation to reality of robot control task, which is an important baseline in the study of domain randomization and is also the baseline algorithm in [Mehta et al., 2020]. Mehta et al. claimed that uniform sampling environment parameters is suboptimal and propose active domain randomization to optimize the sampling distribution, whereas VRDR aims to reduce the variance of gradient estimate. We argue that VRDR are applicable to advanced domain randomization of optimized sampling distribution. We have proved that for arbitrary sampling distribution of environment parameters, if Eq. (11) holds, we still have $Var^{b^*(s)}(g) \geq  Var^{M}(g)$.  Namely, VRDR can still reduce variance of gradient estiamte.  We will provide the detail proof in the revised version. Also, we are now conducting experiments to evaluate the active domain randomization algorithm in our robot simulation environments and the results will be provided later.

---

> > > ### Comment · Reviewer_kFGn · 2021-11-15
> > > **Thm 1**
> > >
> > > You cannot cancel the G term in Equation (3) so the derivation does not go through. Could you show me how this equation could hold?

---

> > > > ### Author Response · Authors · 2021-11-16
> > > > **Response to Thm1**
> > > >
> > > > **Comment 8**: "You cannot cancel the G term in Equation (3) so the derivation does not go through. Could you show me how this equation could hold?"
> > > >
> > > > **Response**:Thanks for your comment. This concern might be addressed by the following two points.
> > > > 1) The G term was not canceled in Eq. (3). Instead, we put  $\frac{G(a,s)}{ \mathbb{E}_{p\sim P,s \sim \mu^p_\pi,a \sim \pi(a|s)} [G(a,s)] }$ into the definition of $V'(s|p)$ as in the third line below Eq. (3). By doing so, we intended to show one possible explanation on the optimal constant baseline $b^*_c$, i.e., interpreted as the expected state value function $V’(s|p)$ over all possible states and environments. And the detailed derivation of the second equality in Eq. (3) is:
> > > >
> > > >
> > > > $$
> > > > 	b_c^* = \frac{E_{P,\mu_\pi^p,\pi} \big [G(a,s)Q_\pi(s,a\vert p) \big ]}{E_{P,\mu_\pi^p,\pi}  \big [G(a, s) \big]} = \frac{E_{P,\mu_{\pi}^p} \biggl [ E_\pi \big [G(a,s)Q_\pi(s,a\vert p) \big ] \biggr ]}{E_{P,\mu_\pi^p,\pi} \big [G(a, s) \big]} =E_{P,\mu_\pi^p} \left [ E_{\pi}\left [ \frac{G(a,s)Q_\pi(s,a\vert p)}{E_{P,\mu_\pi^p,\pi} \big [G(a, s) \big]} \right ] \right ] \triangleq E_{P,\mu_\pi^p} \big [V'(s\vert p) \big]
> > > > $$
> > > > Here $E$ is used to represent $\mathbb{E}$.
> > > >
> > > > 2) The derivation of Theorem 1 was not based on Eq. (3).  Theorem 1 quantifies the variance reduction improvement of gradient estimator by incorporating the optimal state/environment-dependent baseline $b^*(s, \mathcal{P})$ (as presented in Eq. (6)) over an arbitrary state-dependent baseline $b(s)$. For the detailed derivation of Theorem 1, please refer to Appendix A.2, where the derivation is not related to Eq. (3).

---

> > > > > ### Comment · Reviewer_kFGn · 2021-11-19
> > > > > **The authors are right**
> > > > >
> > > > > I somehow mis-interchanged (3) and (6). Indeed, when the baseline additionally involves the environment this marginal gain of variance reduction is guaranteed (compared to the state-dependent baseline). I take back comment 1. Thank the authors for pointing out.

---

> ### Author Response · Authors · 2021-11-23
> **Final Response and Summary of Changes to Reviewer KFGn**
>
> We would like to thank again the reviewer for providing the detailed comments and discussing with us on the theoretical derivations. To address these comments, we have also made corresponding changes, as highlighted in blue, in the rebuttal version of our paper. Here, we would like to summarize the major revisions that we have made in the rebuttal version.
>
> **1.	Evaluation of variance of gradient estimates during training**: We have provided the practical training curves, presenting the variance of gradient estimates of DR, MVN and VRDR during training on Pendulum2D in Fig. 10. It can be seen from Fig. 10 that the beginning of training, VRDR can achieve a significantly lower variance of gradient estimates than the other two baselines, which coincides with the faster convergence behavior of VRDR as shown in Fig. 1(e).
>
> **2.	Additional computational cost of VRDR**: We have provided complexity analysis of VRDR in Appendix A.8, showing that the additional computational cost introduced by the clustering process is not the dominate term, compared to the feed-forward and back-propagation computational cost introduced by the policy and value networks in DR.
>
> **3.	Evaluation of other clustering method in VRDR**: We have conducted experiments to evaluate using k-means as the clustering method in VRDR instead of our original choice of hierarchical clustering. The training curves have been provided in Fig. 6(a) and Fig. 6(b) in Appendix A.10, which present similar average scores on Hopper and Halfcheetah.
>
> **4.	Clarity of Algorithm 1**: We have defined parameter Nc at the beginning of Algorithm 1.
>
> **5.	Clarity of Section 5.2**: We have clarified the description of “15 policies” at the same place in Section 5.2.
>
> **6.	Uniform sampling DR and comparison to ADR**: Due to limit of our computing equipment, the evaluation of ADR is unfortunately not finished. Instead, we have provided derivation in Appendix A.6 to show that Theorem 2 is valid for and VRDR can also be applied to the general DR with an arbitrary sampling distribution for the environment parameters.
>
> **7.	Other concerns**: Please refer to our initial responses to Comments 1, 2 and 8.

---

> > ### Comment · Reviewer_kFGn · 2021-11-24
> > **Thanks for the response**
> >
> > I have noted the updates the authors have made and I thank the authors for the clarification and for improving the manuscript.

---

### Official Review · Reviewer_LpEU · 2021-11-04

**Correctness:** 4
**Technical Novelty And Significance:** 2
**Empirical Novelty And Significance:** 2
**Recommendation:** 5
**Confidence:** 4

**Main Review:**

The paper is well written and the mathematical derivation is sound. The idea of extending the state values as baselines to the additional parameterization of the environment variations is natural. Because the variance reduction technique in standard single-environment training is well-studied, the extension to the additional parameterization is relatively straightforward (the technical details are well-presented).

My main concerns are about the experimental results: I think the analysis is too weak and not enough baselines are compared with. By looking at Figure 1, it is quite hard to conclude that the proposed methods really help (whereas the standard variance reduction does make a crucial difference in training). I hope the authors delve more into the results and in particular elaborate on possible reasons when the benefits are unclear, rather than just a simple paragraph on a few environments where the method is marginally better (and I would guess there can be some examples where it is worse?). I do not believe these are all the baselines that should be considered, for instance there's no comparison with meta learning or robust RL methods.

**Summary Of The Paper:**

The paper derived an optimal state/environment-dependent baseline and a variance reduced domain randomization approach for policy gradient methods.

**Summary Of The Review:**

The paper is well written and the mathematical derivation is sound. My main concerns are about the experimental results: I think the analysis is too weak and not enough baselines are compared with.

---

> ### Author Response · Authors · 2021-11-15
> **Initial Response to Reviewer LpEU**
>
> We would like to thank the reviewer for providing the detailed comments. Here, we just wanted to provide our initial response to these comments, and possibly discuss with the reviewer about our revision plan. The final formal response and revised manuscript will be submitted at the end of the discussion period.
>
> **Comment 1**: “I hope the authors delve more into the results and in particular elaborate on possible reasons when the benefits are unclear, rather than just a simple paragraph on a few environments where the method is marginally better (and I would guess there can be some examples where it is worse?)”
>
> **Response**: Thanks for this suggestion. We are currently conducting experiments to evaluate the variance of the gradient estimator of DR, MVN and VRDR, and trying to build up an empirically justified relation between the variance of gradient estimator and algorithm’s convergence performance. The results will be provided later when the experiments are done.
>
> **Comment 2**: “I do not believe these are all the baselines that should be considered, for instance there's no comparison with meta learning or robust RL methods.”
>
> **Response**: We are now conducting experiments of evaluating other baseline algorithms including active domain randomization (Mehta), EPOPT (Rajeswaran) and meta input-driven baseline (Mao) and the results will be provided later in the revised version.
>
> **References**:
> Bhairav Mehta, Manfred Diaz, Florian Golemo, Christopher J Pal, and Liam Paull. Active domain randomization. In Conference on Robot Learning, pp. 1162–1176. PMLR, 2020.
>
> Rajeswaran A, Ghotra S, Ravindran B, et al. Epopt: Learning robust neural network policies using model ensembles[J]. arXiv preprint arXiv:1610.01283, 2016.
>
> Hongzi Mao, Shaileshh Bojja Venkatakrishnan, Malte Schwarzkopf, and Mohammad Alizadeh.
> Variance reduction for reinforcement learning in input-driven environments. In International
> Conference on Learning Representations, 2018.

---

> ### Author Response · Authors · 2021-11-23
> **Final Response and Summary of Changes to Reviewer LpEU**
>
> We would like to thank again the reviewer for providing the detailed comments. To address these comments, we have also made corresponding changes, as highlighted in blue, in the rebuttal version of our paper. Here, we would like to summarize the major revisions that we have made in the rebuttal version.
>
> **1.	Reasons of VRDR’s faster convergence**: In the rebuttal version, we have empirically evaluated the variance of gradient estimates during training in Appendix A.13. Specifically, we have provided the practical training curves, presenting the variance of gradient estimates of DR, MVN and VRDR during training on Pendulum2D in Fig. 10. It can be seen from Fig. 10 that the beginning of training, VRDR can achieve a significantly lower variance of gradient estimates than the other two baselines, which coincides with the faster convergence behavior of VRDR as shown in Fig. 1(e).
>
> **2.	Comparison to other baseline algorithms**: We have conducted experiments to evaluate other baseline algorithms, including DR with meta baseline (DRMB) and Epopt (a robust RL baseline). The results have been provided in Figs. 8(a)-8(b) and Figs. 9(a)-9(b), respectively, in Appendix A.12. It can be seen that in most of the cases, VRDR achieves a better performance, in terms of the convergence speed and the average return.

---

### Official Review · Reviewer_ePpB · 2021-11-08

**Correctness:** 3
**Technical Novelty And Significance:** 4
**Empirical Novelty And Significance:** 2
**Recommendation:** 5
**Confidence:** 2

**Main Review:**

strengths
1. this work has a good structure, is well-written, and I feel pleased to read this paper
2. this work is well-motivated and I believe the topic is highly-significant and related to the venue
3. this work has a good balance of theoretical contribution and empirical evaluation. it is technically novel and the proposed algorithm works well in practice compared to baselines

weaknesses/questions
1. this work targets the variance of the gradient estimates. so why not directly measure the variance of the gradient estimates? e.g. empirically? I think this is the most straightforward way.
2. I agree lower variance gradient estimates could converge fast, but in general, it's also easier to converge to local optimal. what are your opinions on this?
4. for sec.5.2, could you elaborate more on why VRDR generalizes better to unseen environments? I don't find it intuitive to understand why lower-variance gradient estimates find a solution that generalizes better (as opposed to sgd in supervised learning)? what's the connection or insight?
5. what's MRPO at the end of sec.5.1? probably a typo?
6. could you elaborate on the added computational cost compared to the baselines?

**Summary Of The Paper:**

this work studies policy gradient methods for domain randomization (DR). in particular, it investigates baselines for policy gradient under the dr settings such that the gradient estimate can have a lower variance to ensure better policy updates and learning. this paper derives optimal state/environment-dependent baseline theoretically, gives general recipes for building the baseline, and proposes an algorithm called variance reduced domain randomization (VRDR). vrdr is evaluated on several continuous control tasks and it performs better compared with two baselines.

**Summary Of The Review:**

I enjoy reading the paper and I find it theoretically novel and sound. However, I do not find the experiments very straightforward, I believe it's necessary to compare the empirical variance of the gradient estimates with the baselines.

---

> ### Author Response · Authors · 2021-11-15
> **Initial Response to Reviewer ePpB**
>
> We would like to thank the reviewer for providing the detailed comments. Here, we just wanted to provide our initial response to these comments, and possibly discuss with the reviewer about our revision plan. The final formal response and revised manuscript will be submitted at the end of the discussion period.
>
> **Comment 1**: “this work targets the variance of the gradient estimates. so why not directly measure the variance of the gradient estimates? e.g. empirically? I think this is the most straightforward way.”
>
> **Response**:  Thanks for this suggestion. We are currently conducting experiments to evaluate the variance of the gradient estimator of DR, MVN and VRDR. The results will be provided later when the experiments are done.
>
> **Comment 2**: “I agree lower variance gradient estimates could converge fast, but in general, it's also easier to converge to local optimal. what are your opinions on this?”
>
> **Response**: In our opinions, lower variance policy gradient estimator $\frac{1}{N_g}\sum g_i$ will be closer to the true policy gradient $\nabla_{\theta}\mathbb{E}[\eta(\pi \vert p)]$, as compared to higher variance estimator, which has a higher (instead of lower) chance to deviate from the direction of maximizing $\mathbb{E}_{p \sim P}[\eta(\pi \vert p)]$ and stuck in the local optimum.
>
> **Comment 3**: “for sec.5.2, could you elaborate more on why VRDR generalizes better to unseen environments? I don't find it intuitive to understand why lower-variance gradient estimates find a solution that generalizes better (as opposed to sgd in supervised learning)? what's the connection or insight?”
>
> **Response**: VRDR aims to reduce the variance of gradient estimate under DR setting, and thus can achieve higher converged average score or faster convergence rate, or both of them. As we have stated in Section 5.2, the better generalization performance can be attained when higher converged average score is achieved, such as in Walker2D, where VRDR shows better converged score than DR and MVN and hence gets better performance on unseen environments as shown in Figure 2 (a). In comparison, faster convergence without a higher converged average score does not guarantee the generalization performance. Like in Pendulum2D, though VRDR converges the fastest, all the three algorithms achieve similar converged average score, and hence similar generalization performance is observed in Figure 2 (b).
>
> **Comment 4**: “what's MRPO at the end of sec.5.1? probably a typo?”
>
> **Response**: We apologize for this typo, and will correct it to VRDR in the revised version.
>
> **Comment 5**: “could you elaborate on the added computational cost compared to the baselines?”
>
> **Response**: Compared to MVN, the added computational cost of VRDR mainly comes from the clustering computation. Compared to DR, both MVN and VRDR have additional computational cost on maintaining the multi-value function. We are now carrying out a derivation on algorithm complexity analysis and will provide it later.

---

> ### Author Response · Authors · 2021-11-23
> **Final Response and Summary of Changes to Reviewer ePpB**
>
> We would like to thank again the reviewer for providing the detailed comments. To address these comments, we have also made corresponding changes, as highlighted in blue, in the rebuttal version of our paper. Here, we would like to summarize the major revisions that we have made in the rebuttal version.
>
> **1.	Evaluation of variance of gradient estimates during training**：We have provided the practical training curves, presenting the variance of gradient estimates of DR, MVN and VRDR during training on Pendulum2D in Fig. 10 in Appendix A.13.
>
> **2.	Additional computational cost of VRDR**：We have provided complexity analysis of VRDR in Appendix A.8.
>
> **3.	Typo**： We have corrected MRPO to VRDR in Section 5.1 of the rebuttal version.
>
> **4.	Other concerns**： Please refer to our initial responses to Comments 2 and 3.

---

> > ### Comment · Reviewer_ePpB · 2021-11-29
> > **thank you for the detailed replys and still some questions regarding the gradient estimates**
> >
> > I would like to thank the authors for providing helpful replies. And I just have a few questions regarding the gradient estimates.
> >
> > 1. Could you elaborate more on how you actually empirically calculate the variance of the gradient estimate?
> > 2. Is the results the same as in the experiment? i.e. from 15 seeds?
> > 3. If yes on 2, could you also plot the confidence bars? It should be easy if you already have the results for 15 seeds, or you have saved the models during learning for the experiments. The reason for that is not only the mean is important, but also the variance.
> > 4. Any particular reason for choosing Pendulum2D? Except for computational reason (if this is the reason then Pendulum and InvertedDoublePendulum should be similar I guess)? The reason I am interested in is that Pendulum2D is not the one VRDR outperforms the others the most in terms of either convergence (Pendulum is faster) or final performance (Walker2D)? So it's necessary and reasonable to understand what brings the performance in Pendulum or Walker2D. And is the low variance the main reason?
> > 5. And in fig. 10, in later stages, VRDR has similar if not higher variance. Could you elaborate what's your opinions on that?

---

> > > ### Author Response · Authors · 2021-11-30
> > > **Response to Reviewer ePpB regarding the Gradient Estimate Part 1**
> > >
> > > We would like to thank the reviewer for the detailed comments.
> > >
> > > **Comment 6**: "Could you elaborate more on how you actually empirically calculate the variance of the gradient estimate?"
> > >
> > > **Response**: We aimed to reduce the variance of gradient estimate for the state-action pair $g(\theta, s,a, p) = \nabla_{\theta} \log \pi_{\theta} (a \vert s) [Q_\pi(s,a,p)-b]$ under each environment $p$, where $\theta$ is the parameter of policy $\pi_{\theta}$. In definition, the variance of $g(\theta, s,a, p)$ can be obtained as
> > > \begin{align}
> > >     Var(g) = E[g(\theta, s,a, p) - E(g(\theta, s,a, p))]^{\mathrm{T}}E[g(\theta, s,a, p) - E(g(\theta, s,a, p))],
> > > \end{align}
> > > where the expectation is taken over $p \sim P$, $s \sim \mu_{\pi}^p$ and $a \sim \pi_{\theta}$.
> > >
> > > Empirically, we estimated $Var(g)$ based on the samples of environment parameter $p$, state $s$ and action $a$. Specifically, at each iteration $k$ where the policy $\pi_{\theta_k}(a \vert s)$ (i.e., the parameter $\theta_k$) is fixed, we sampled $H$ environments {$p_c$}$_{c=0}^{H-1}$ with $L$ trajectories for each environment $p_c$. We then calculated the $g(\theta_k, s,a, p_c)$ value for each state-action pair $(s,a)$ in the sampled environment $p_c$ based on these trajectories, while $E[g(\theta_k, s,a, p_c)]$ was estimated by taking the average of $g(\theta_k, s,a, p_c)$ for the same tuple of $(s,a, p_c)$. The variance of $g(\theta_k, s,a, p_c)$ could then be obtained by applying the empirical estimate of above definition equation `var = mean(abs(g - g.mean())**2)` , which was shown in Fig. 10 as the mean variance.
> > >
> > > **Comment 7**: "Is the results the same as in the experiment? i.e. from 15 seeds?"
> > >
> > > **Response**: According to the above-described empirical calculation of variance of the gradient estimate, it takes a very high computational cost to calculate $g(\theta_k, s,a, p_c)$ for each tuple of $(s,a, p_c)$. Due to the time constraint, unfortunately before the rebuttal deadline we only managed to finish the experiment on Pendulum2D with one seed.
> > >
> > > **Comment 8**: If yes on 2, could you also plot the confidence bars? It should be easy if you already have the results for 15 seeds, or you have saved the models during learning for the experiments. The reason for that is not only the mean is important, but also the variance.
> > >
> > > **Response**: Continuing with the Response to Comment 7, we apologize for the mistake in the vertical axis label, which should be changed from "mean variance" to "variance" to be more precise. While the text in Appendix A.13 does not need to change, where we used the precise term "variance of the gradient estimates".
> > >
> > > **Comment 9**: Any particular reason for choosing Pendulum2D? Except for computational reason (if this is the reason then Pendulum and InvertedDoublePendulum should be similar I guess)? The reason I am interested in is that Pendulum2D is not the one VRDR outperforms the others the most in terms of either convergence (Pendulum is faster) or final performance (Walker2D)? So it's necessary and reasonable to understand what brings the performance in Pendulum or Walker2D. And is the low variance the main reason?
> > >
> > > **Response**: Still referring to the Response to Comment 7, the reason for choosing Pendulum2D is mainly because of the high computational cost of empirical gradient calculation and the limit of our computing resource. Within all the six tasks, Pendulum2D and Pendulum both required the minimum number of iterations to converge (i.e., about 200 iterations). The reason why we chose Pendulum2D, other than Pendulum, is that we thought VRDR already presented a superiority in terms the convergence speed in Pendulum2D and that Pendulum2D is a more complicated task than Pendulum.
> > >
> > >
> > > To relate the variance of gradient estimates to the convergence behavior of policy gradient methods, we provided in Theorem 3 in Appendix A.7 a theoretical analysis of the convergence upper bound, the dominant term $\frac{L \max_k \sqrt{Var(g_k)}}{\sqrt{K}}$ of which is determined by the variance of gradient estimates. This theoretical convergence upper bound implies that by reducing the variance $Var(g_k)$ the convergence will be accelerated. We then showed that compared to the optimal state-dependent baseline $b^*(s)$, our proposed VRDR gained a $\kappa$ acceleration on the convergence of policy gradient methods, where $\kappa^2$ is the ratio of $Var^{b^*(s)}(g)$/ $Var^{M}(g)$.  According to Theorem 2, this acceleration factor $\kappa>1$ would hold as long as the condition in Eq. (11) is satisfied.

---

> > > ### Author Response · Authors · 2021-11-30
> > > **Response to Reviewer ePpB regarding the Gradient Estimate Part 2**
> > >
> > > **Comment 10**: "And in fig. 10, in later stages, VRDR has similar if not higher variance. Could you elaborate what's your opinions on that?"
> > >
> > > **Response**: Although VRDR did not show the best performance in Pendulum2D, we validated that the curve of variance is consistent with the curve of its average score during training. Specifically, in Fig. 10, VRDR showed an advantage of lower variance at the early stage (i.e., before about 180 iterations), while VRDR achieved the fastest convergence rate before almost the same number of iterations as shown in Fig. 1(e). Then in the later stage (after about 180 iterations, DR in Fig. 10 achieved a generally lower variance, which also corresponded to a slightly faster convergence speed of DR as shown in Fig. 1(e). Therefore, the correspondence between the empirical evaluation of the training curve of Pendulum2D in Fig. 1(e) and its variance curve in Fig. 10 also validated our theoretical analysis in Theorem 3. About the reason for VRDR's higher variance in the later stage, we hypothesized that it might because the hyperparameter $N_c$ (i.e., the clustering interval) was kept as a constant in the experiment of Fig. 10, which however should have been adaptively increased in the later stage when the average score approached the convergence to make the condition in Eq. (11) hold.

---

### Author Response · Authors · 2021-11-23
**Summary of Changes in the Rebuttal Version**

We are greatly appreciated for all the anonymous reviewers, for their highly constructive comments to help us to improve the paper. We have taken careful consideration of them, and tried our best to respond to each of them and discuss with the reviewers during the discussion phase.

To address these comments, we have also made corresponding changes, as highlighted in blue, in the rebuttal version of our paper. Here, we would like to summarize the major revisions that we have made in the rebuttal version.



### 1. Concerns on Empirical Evaluations
**1-1) Ablation study of VRDR on clustering methods**:

We have conducted experiments to evaluate using k-means as the clustering method in VRDR instead of our original choice of hierarchical clustering. The training curves have been provided in Fig. 6(a) and Fig. 6(b) in Appendix A.10, which present similar average scores on Hopper and Halfcheetah.

**1-2) Comparison to combined state-dependent baseline**:

We have conducted experiments to evaluate DR with combined state-dependent baseline (DRCS), where environment parameters are treated as extra state of the MDP and the corresponding value function takes combined state (original state and environment parameters) as input. The training curves have been provided in Fig. 7(a) and Fig. 7(b) in Appendix A.11, which show that DRCS degrades the average score during training as compared to DR and VRDR.

**1-3) Comparison to other baseline algorithms**:

We have conducted experiments to evaluate other baseline algorithms, including DR with meta baseline (DRMB) and Epopt. The results have been provided in Figs. 8(a)-8(b) and Figs. 9(a)-9(b), respectively, in Appendix A.12.

**1-4) Evaluation of variance of gradient estimates during training**:

We have provided the practical training curves, presenting the variance of gradient estimates during training on Pendulum2D in Fig. 10 in Appendix A.13.

### 2. Concerns on the Detail of Algorithm 1

**2-1) Clarity of Algorithm 1**:

We have modified Algorithm 1 for more clarity and added Algorithm 2 to describe the hierarchical clustering methods that we have used in practice to implement Algorithm 1. Besides, we have added explanation on how the baselines are updated and the relabeling in Section 4 under Algorithm 1.

**2-2) Additional computational cost**:

We have provided complexity analysis of VRDR in Appendix A.8.

### 3. Concerns on the Computation of E_P[Y_P(S)]:

**3-1) Optimal State-Dependent Baseline in Eq. (4)**:

We have re-derived the optimal state-dependent baseline in Appendix A.1, where Eq. (4) has been corrected from taking the expectation over P(p) to taking the expectation over P(p|s).

**3-2) Variance Reduction Improvement in Corollary 1**:

We have corrected E_P[Y_P(S)] to E_{P(p|s)}[Y_P(S)].

**3-3) Theorem 2 and Its Proof**:

We have provided a more clear derivation of Theorem 2 in Appendix A.5 based on the corrected Eq. (4).

### 4. Concerns on the Uniform Sampling DR:
We have provided derivation in Appendix A.6 to show that Theorem 2 is valid for and VRDR can be applied to the general DR with an arbitrary sampling distribution.

---

### Decision · Program_Chairs · 2022-01-20

**Decision:**

Reject

**Comment:**

While the reviewers appreciated the clarity of the work, there is a concern about the meaning of the proposed result and method. It is known that adding knowledge about an additional variable, in this case the environment, leads to a lower variance estimate. What is not known is the practical impact of using this new baseline or perhaps some other intuition stemming from that use of the baseline (for instance the origin of the variance). However, the results shown are not that compelling, a point which was raised by the reviewers, making the work below the bar for publication.